

# Water isotopic characterisation of the cloud-circulation coupling in the North Atlantic trades. Part 1: A process-oriented evaluation of COSMO$_{iso}$ simulations with EUREC[4]A observations

Leonie Villiger[1,2], Marina Dütsch[3], Sandrine Bony[4], Marie Lothon[5], Stephan Pfahl[6], Heini Wernli[1], Pierre-Etienne Brilouet[7], Patrick Chazette[8], Pierre Coutris[9], Julien Delanoë[10], Cyrille Flamant[11], Alfons Schwarzenboeck[9], Martin Werner[12], and Franziska Aemisegger[1]

[1]Institute for Atmospheric and Climate Science, ETH Zurich, Zurich, Switzerland
[2]Institute for Environmental Decisions, ETH Zurich, Zurich, Switzerland
[3]Department of Meteorology and Geophysics, University of Vienna, Vienna, Austria
[4]LMD/IPSL, Sorbonne Université, CNRS, Paris, France
[5]Laboratoire d'Aérologie, University of Toulouse, CNRS, Toulouse, France
[6]Institute of Meteorology, Freie Universität Berlin, Berlin, Germany
[7]CNRM, Université de Toulouse, Météo-France, CNRS, Toulouse, France
[8]LSCE/IPSL, CNRS-CEA-UVSQ, University Paris-Saclay, Gif sur Yvette, France
[9]LAMP, Université Clermont Auvergne, CNRS, Clermont-Ferrand, France
[10]LATMOS/IPSL, Université Paris-Saclay, UVSQ, Guyancourt, France
[11]LATMOS/IPSL, Sorbonne Université, CNRS, Paris, France
[12]Alfred Wegener Institute Helmholtz Centre for Polar and Marine Research, Bremerhaven, Germany

**Correspondence:** Leonie Villiger (leonie.villiger@env.ethz.ch)

**Abstract.** Stable water isotope observations have the potential to provide information on cloud processes in the trade-wind region, in particular when combined with high-resolution model simulations. In order to evaluate this potential, nested convection-resolving COSMO$_{iso}$ simulations with horizontal grid spacings of 10, 5, and 1 km were carried out in this study over the tropical Atlantic for the time period of the EUREC[4]A field experiment. To keep the conditions in the domain as close as possible to the real meteorology, we applied a spectral nudging of horizontal winds towards reanalysis data. The comparison to airborne in situ and remote sensing observations shows that the three simulations are able to distinguish between different mesoscale cloud organisation patterns as well as between periods with comparably high and low rain rates. Precipitation, cloud fraction and liquid water content are sensitive to the grid spacing. Cloud fraction and liquid water content show a better agreement with aircraft observations with higher spatial resolution. Contrastingly, temperature, humidity, and isotopes in vapour remain fairly unaffected by the model resolution. A low-level cold-dry bias, including too depleted vapour in the subcloud and cloud layer and too enriched vapour in the free troposphere, is found in all three simulations. Furthermore, the simulated secondary isotope variable d-excess in vapour is overestimated compared to observations. Special attention is given to the cloud base level, the formation altitude of shallow cumulus clouds, which are rooted in the thermals of the subcloud layer. The temporal variability of the simulated isotope variables at cloud base agrees reasonably well with observations, with correlations of the flight-to-flight data as high as 0.69 for $\delta^2$H and 0.74 for d-excess. A close examination of different mesoscale cloud base features, including clouds and clear-sky dry-warm patches, and their isotopic characteristics shows that i) these features are





represented faithfully in the model with similar frequency of occurrence, isotope signals and specific humidity anomalies as found in the observations ($+2\,‰$ to $+5\,‰$ [$+2\,\mathrm{g\,kg^{-1}}$] for precipitating clouds vs. $-3\,‰$ to $-4\,‰$ [$-2\,\mathrm{g\,kg^{-1}}$] in dry-warm patches for $\delta^2\mathrm{H}$ [$q$]) and ii) the $\delta^2\mathrm{H}$ of cloud base vapour at the hourly time scale is mainly controlled by mesoscale transport and not by local microphysical processes while the d-excess is mainly controlled by large-scale drivers. Overall, this evaluation of COSMO$_{\mathrm{iso}}$, including the isotopic characterisation of cloud base features, suggests that the simulations can be used for investigating the role of atmospheric circulations on different scales for controlling the formation of shallow cumulus clouds in the trade-wind region, as will be done in part 2 of this study. Additionally, we provide explicit recommendations for adaptations of the modelling setup to be tested in future research.

## 1 Introduction

Shallow clouds, ubiquitous in the trade-wind region, substantially contribute to the cooling of the Earth's climate through their shortwave radiative effect. Their response to climate change is unclear, contributing to a large part of the uncertainty of climate projections (e.g., Bony and Dufresne, 2005; Zelinka et al., 2017; Schneider et al., 2017). The cloud fraction at cloud base, in particular, has been identified as a key parameter for the spread of the modelled feedback of these clouds to climate change (Bony et al., 2017). Therefore, understanding the processes controlling the variability of cloudiness at cloud base is of utmost importance.

The field campaign EUREC$^4$A (Elucidating the role of clouds-circulation coupling in climate; Stevens et al., 2021), which took place in early 2020 in the vicinity of the Caribbean island Barbados, was designed to provide observational constraints of the mechanisms that control shallow trade-wind clouds. A focus of the campaign was the mesoscale organisation pattern of these clouds, with the most frequent four patterns named Sugar, Gravel, Flower, and Fish (Stevens et al., 2020; Bony et al., 2020; Schulz, 2022). As part of EUREC$^4$A, the EUREC$^4$A$_{\mathrm{iso}}$ component coordinated a multi-platform network of stable water isotope measurements (Bailey et al., 2023). Stable water isotopologues (hereafter simply named stable water isotopes) reflect the integral of moist processes experienced by an air parcel during transport (Gat, 1996; Galewsky et al., 2016). Hence, they are a promising tool for bridging the gap between microphysical processes at the scale of clouds and transport processes on larger scales.

Here, we use the isotope measurements that were conducted onboard the French aircraft SAFIRE ATR-42 (hereafter ATR; Bony et al., 2022), whose mission during EUREC$^4$A was to provide a detailed characterisation of atmospheric properties near the cloud-base level and within the subcloud layer. The measurements, which are limited in space and time, are complemented with three numerical simulations that were performed using the isotope-enabled version of the non-hydrostatic weather forecast model from the Consortium for Small Scale Modelling COSMO$_{\mathrm{iso}}$ (Steppeler et al., 2003; Pfahl et al., 2012). Providing a description of the three-dimensional distribution of isotopes at hourly intervals, the simulations can be used to test hypotheses based on the observational data. However, it is important to evaluate the COSMO$_{\mathrm{iso}}$ simulation data before using it to deepen our process understanding. Three objectives are pursued: (1) evaluate the ability of COSMO$_{\mathrm{iso}}$ to reproduce key characteristics of shallow cumulus clouds in the trade-wind region, (2) assess the effect of the spatial model resolution on different variables





characterising the occurrence and the environment of shallow trade-wind cumulus clouds, and (3) conduct a feature-based assessment (including cloudy and clear-sky patches) of isotope variability at cloud base, where shallow cumulus cloud formation is initiated.

This section continues with a short introduction into stable water isotope physics, a general description of the trade-wind boundary layer, and a summary of the results from earlier, relevant model evaluation studies. After the introduction (Sect. 1),
the applied datasets and methods are described (Sect. 2), followed by the results of the evaluation of the three COSMO$_{\mathrm{iso}}$ simulations (Sect. 3) and the assessment of the isotope variability at cloud base (Sect. 4). The paper ends with a summary of the most important findings and their consequences for investigating the role of mesoscale circulations for shallow cumulus cloud formation using water isotopes as tracers (Sect. 5).

### 1.1 Stable water isotopes

Heavy stable water isotopologues (isotopes, hereafter) are water molecules containing a heavy hydrogen (i.e., $^{1}\mathrm{H}^{2}\mathrm{H}^{16}\mathrm{O}$) or oxygen atom (i.e., $^{1}\mathrm{H}_{2}^{18}\mathrm{O}$). As a result, they have lower saturation vapour pressures and lower diffusion velocities than their light counterpart (i.e., $^{1}\mathrm{H}_{2}^{16}\mathrm{O}$), which leads to a change in the relative abundance of heavy to light isotopes during phase transitions. The isotopic composition of a water sample is typically communicated with the so-called $\delta$-notation for $^{2}\mathrm{H}$ and $^{18}\mathrm{O}$, respectively:

$$\delta^{2}\mathrm{H}\,[‰] = \left( \frac{^{2}R_{sample}}{^{2}R_{VSMOW}} - 1 \right) \cdot 1000 \tag{1}$$

$$\delta^{18}\mathrm{O}\,[‰] = \left( \frac{^{18}R_{sample}}{^{18}R_{VSMOW}} - 1 \right) \cdot 1000 \tag{2}$$

The $R$ in Eq. 1 and Eq. 2 stands for the atomic ratio of the concentration of the heavy (rare) to the light (most abundant) isotope, i.e., $^{2}R = \frac{[^{2}\mathrm{H}]}{[^{1}\mathrm{H}]}$ and $^{18}R = \frac{[^{18}\mathrm{O}]}{[^{16}\mathrm{O}]}$, in the water sample and the internationally accepted Vienna Mean Ocean Standard Water 2
(VSMOW; International Atomic Energy Agency, 2017). Samples with high $\delta^{2}\mathrm{H}$ and $\delta^{18}\mathrm{O}$ values are referred to as enriched (in heavy isotopes), and samples with low $\delta^{2}\mathrm{H}$ and $\delta^{18}\mathrm{O}$ values are referred to as depleted (in heavy isotopes). The relative variations of $\delta^{2}\mathrm{H}$ and $\delta^{18}\mathrm{O}$ are assessed with the deuterium excess (d-excess), defined as $d = \delta^{2}\mathrm{H} - 8 \times \delta^{18}\mathrm{O}$, which is a measure for the thermodynamic disequilibrium of the environment during phase transitions (e.g., Gat, 1996; Pfahl and Wernli, 2008).

The usefulness of isotopes has been demonstrated in numerous earlier studies. For example, it is known that extratropical cyclones and cold front passages leave a clear and attributable signal in surface isotope measurements (e.g., Aemisegger et al., 2015; Aemisegger, 2018; Graf et al., 2019; Thurnherr et al., 2020). In the (sub)tropics, isotope observations could be linked to the large-scale circulation (Torri et al., 2017; Guilpart et al., 2017), the daily cycle of the boundary layer (Noone et al., 2011), and precipitation efficiency in shallow convection (Bailey et al., 2015). The links identified in these studies build on the
facts that i) the isotope signal of water vapour is modified during transport and therefore reflects pathway-specific processes, and ii) that the formation altitude of rainwater largely determines its isotopic composition. Airborne measurements have been





considered highly valuable as they provide a three-dimensional description of isotope gradients (Dyroff et al., 2015; Sodemann et al., 2017; Salmon et al., 2019; Chazette et al., 2021), which, for instance, allows to investigate vertical mixing processes. Furthermore, isotope-enabled models have been shown to be useful in looking into tropical deep convection (Bony et al., 2008;

Risi et al., 2008; Blossey et al., 2010; Risi et al., 2020, 2021; de Vries et al., 2022), entrainment of free-tropospheric air into the tropical marine boundary layer (Risi et al., 2019), or precipitation-driven cold pool dynamics (Torri and Kuang, 2016a, b; Torri, 2021).

## 1.2 The trade-wind boundary layer

In this study, we focus on the isotopic characteristics of the lower troposphere in the trade-wind region, which is typically

divided into the subcloud layer, the cloud layer, and the free troposphere (Fig. 1; see also Wood, 2012, in which many of the relevant processes described here are addressed in more detail). The subcloud layer and the cloud layer make up the boundary layer. The subcloud layer connects the cloud layer with the ocean surface, providing the buoyancy and moisture for the formation of new clouds. Often it is assumed to be a well-mixed layer with negligible vertical gradients in humidity and temperature. Recently, however, Albright et al. (2022, their Fig. 2 and 4) showed that the well-mixed layer (its top being

characterised by a jump in specific humidity, potential temperature, or relative humidity) does not always extend to the subcloud layer top (identified via buoyancy measures and corresponding to the lifting condensation level in measurements), but is sometimes separated from it by a thin transition layer. Furthermore, they found that the subcloud layer depth and fluxes are largely controlled by near-surface wind speed, while moisture and heat within the subcloud layer are more sensitive to the thermodynamic conditions above the subcloud layer top. Note that the occurrence of cold pools can modify this picture,

generating a smaller cooler mixed layer overlaid by a stable layer (Touzé-Peiffer et al., 2022).

Directly adjunct to the subcloud layer top is the cloud base level, where clouds start to form and may grow vertically until reaching the trade inversion, which represents the upper bound of the cloud layer. The horizontal extent of the clouds is typically largest at cloud base, but, if the clouds develop up to the trade inversion level, a secondary maximum may emerge at their top in the form of stratiform outflows. This leads to a bi-modal vertical distribution of cloud fraction (Vial et al., 2019, e.g.,). The

contrasting radiative properties of the cloudy and clear-sky environments in the cloud layer can induce shallow circulations (Naumann et al., 2017), which in turn influence the properties of the subcloud layer below. Another process through which the cloud layer feeds back to the subcloud layer is the formation of evaporatively driven cold pools, which create cold-dry density currents that spread radially at the surface. They may initiate new convection at their edges through mechanical lifting and buoyancy caused by increased moisture contents (e.g., Tompkins, 2001; Schlemmer and Hohenegger, 2014; Torri et al., 2015;

Vogel et al., 2021).

The trade inversion that caps the cloud layer forms due to the large-scale subsidence typically found in the free troposphere of the trade-wind region. This large-scale subsidence corresponds to the descending branch of the Hadley cell. The subsidence rate is determined by the balance between adiabatic warming and radiative cooling and is roughly $35\,\mathrm{hPa\,d^{-1}}$ (Salathé and Hartmann, 1997; Holton and Hakim, 2013). Occasionally, however, the Hadley-cell-like descent is replaced by air masses





descending much faster ($\sim 200\,\mathrm{hPa\,d^{-1}}$) from the extratropics along the slanted isentropes (Aemisegger et al., 2021b; Villiger et al., 2022).

    Each of the above-described layers (Fig. 1) is characterised by a distinct isotopic composition of vapour. Generally, the vapour is more enriched in heavy isotopes closer to the surface. With increasing altitude, as soon as condensation and rainout occur, the vapour continuously loses its heavy isotopes and becomes lighter. The d-excess is assumed to be high near the ocean

surface and to decrease with altitude due to a slight temperature dependency (Thurnherr et al., 2021). Once very dry conditions are reached (at about $12\,\mathrm{km}$ and above; this altitude was identified in vertical cross sections [not shown] of the data from the numerical simulations described in Sect. 2) the non-linearity of the $\delta$-scale causes a rapid increase of the d-excess relative to the values observed near the surface (Dütsch et al., 2017). The large-scale circulation can substantially alter the isotopic composition (Aemisegger et al., 2021b) by influencing the conditions under which the vapour is evaporated from the ocean

surface or by advecting air masses from other regions (e.g., more depleted air from higher latitudes).

    At cloud base, different processes are at play shaping the isotope variability. On the one hand, transport within mesoscale systems accumulates freshly evaporated and enriched vapour in areas where clouds are being formed and brings dry and depleted air from high altitudes to cloud base into the clear-sky environments between clouds (radiatively driven subsidence). The d-excess is expected to be high in convective updrafts bearing the signal of ocean evaporation and comparably lower for

moisture in clear-sky environments, which has previously gone through cloud formation processes (i.e., moisture detrained from clouds). Note that in the shallow trade-wind cumuli regime, the air does not become dry enough for the non-linearity of the $\delta$-scale to affect the d-excess. On the other hand, microphysical processes, such as the formation of cloud droplets or the partial or full evaporation of cloud and rain droplets, impact the cloud vapour's isotope signal. If the formed condensate is removed from the cloud through precipitation, the cloud layer's total water gets depleted in heavy isotopes. The vapour

in clear-sky environments is indirectly influenced by this process, as ultimately, the cloudy air gets detrained into clear-sky environments and subsides back to cloud base. Both processes, mesoscale transport and microphysics, taken together, thus, lead to contrasting isotope signals in cloudy and clear-sky environments at cloud base.

### 1.3   Earlier model evaluation studies

Several studies have evaluated the performance of models in the lower troposphere of the trade-wind region during EUREC[4]A.

Beucher et al. (2022) tested the oversea configuration of the French regional model AROME, called AROME-OM (its output used by Dauhut et al., 2022, as initial and lateral boundary conditions for large eddy simulations). In their model setup, deep convection is treated explicitly, but shallow convection is parameterised. Beucher et al. (2022) found that the AROME-OM produces a too deep subcloud layer associated with a dry bias ($-1\,\mathrm{g\,kg^{-1}}$ in specific humidity) and has a cold bias in the cloud layer ($-0.5\,\mathrm{K}$ in potential temperature). Further, the AROME-OM is able to produce stratiform clouds near the trade inversion,

but their occurrence is underestimated. Besides identifying these shortcomings, Beucher et al. (2022) demonstrated the capacity of the AROME-OM to predict the different mesoscale cloud organisation patterns and their associated environment. Savazzi et al. (2022) looked into lower-tropospheric biases of meridional winds in Integrated Forecasting System (IFS) forecasts and ERA5 reanalyses. They found a weak wind speed bias at altitudes below $5\,\mathrm{km}$ during local day and a strong wind speed bias



below 2 km during local nighttime. Here, a further model evaluation in the framework of EUREC[4]A is added with a focus on

isotopes.

Since we are using COSMO$_{iso}$ simulations, we summarise the most relevant findings from earlier model evaluation studies of COSMO and COSMO$_{iso}$ used at the km-scale resolution, albeit conducted mostly in regions other than the trades. Ban et al. (2014) showed that turning off the deep convection scheme leads to a more accurate diurnal cycle of surface temperature and precipitation over Europe. Vergara-Temprado et al. (2020) performed COSMO simulations over Europe at seven differ-

ent horizontal grid spacings ranging from 50 km to 2.2 km. At each resolution, Vergara-Temprado et al. (2020) carried out one simulation with the full Tiedtke (1989) scheme, one with a parameterisation of shallow convection only, and another one without any convection parameterisation. They found that at grid spacings of 25 km or smaller, explicit convection improves the model skill with regard to surface temperature, precipitation, and top-of-atmosphere radiation. The improvement over the simulations with fully parameterised convection applies to both simulations with partly resolved (deep convection) and fully

resolved convection (deep and shallow). Furthermore, the study showed that the representation of some variables strongly depends on the spatial resolution of the model, while for others, the convection parametrisation settings (i.e., resolved vs. parameterised) are crucial. For example, hourly precipitation and outgoing longwave radiation are mostly determined by whether the convection is resolved or parameterised, while net shortwave radiation is more sensitive to the spatial resolution (due to its strong dependence on cloud fraction). The grid-spacing sensitivity of COSMO was further investigated by Heim et al. (2021).

They carried out five convection-resolving simulations over the tropical Atlantic with a grid spacing ranging from 12 km to 500 m and found, for instance, that the low-cloud fraction decreases with finer resolution.

The isotope-enabled version of COSMO has been used and evaluated previously in different regions, but never in the tropical trade-wind region. Thurnherr et al. (2021) used COSMO$_{iso}$ simulations with explicit convection over the Southern Ocean and compared them to ship-based and radiosonde measurements. They observed that the vapour in the lowest model level was

too depleted and had too high d-excess values compared to in situ measurements (their Fig. 9). They argued that too strong vertical mixing possibly caused the observed bias and that the offset in d-excess could be reduced by using the smooth regime of Merlivat and Jouzel (1979)'s formulation for non-equilibrium fractionation, which we use in this study, instead of the formulation by Pfahl and Wernli (2009) developed with observations from the Mediterranean. Previous studies (e.g., Steen-Larsen et al., 2015; Aemisegger and Sjolte, 2018; Bonne et al., 2019) showed that using the non-equilibrium fractionation

factor corresponding to the smooth regime in the Merlivat and Jouzel (1979) formulation provides the most reliable results.

Dahinden et al. (2021) and de Vries et al. (2022) evaluated COSMO$_{iso}$'s humidity and isotope signals in the free troposphere in the context of the West African monsoon. Dahinden et al. (2021), who used a setup with resolved convection in the vicinity of the Canary Islands, found that COSMO$_{iso}$ is overly moist and enriched at altitudes $\geq 6$ km compared to in situ aircraft observations. However, the day-to-day variability of COSMO$_{iso}$'s humidity and $\delta^2$H agreed well with ground-based remote

sensing observations from Teneriffe (FTIR). de Vries et al. (2022) performed three COSMO$_{iso}$ simulations in the West African monsoon region and tested the effect of parameterised versus explicit convection and different spatial resolutions. Independent of the modelling setup, COSMO$_{iso}$ produced a distinct bias towards too high $\delta^2$H at 4.2 km compared to satellite-based remote sensing observations from IASI. Despite the offset, the temporal evolution of the simulations agreed well with the observations



lending confidence to the model's ability to reproduce the mesoscale to synoptic-scale variability in the water vapour isotope
fields.

The literature summarised above suggests that simulations should be done by using an explicit representation of convection
and for variables such as the cloud fraction a fine spatial resolution (Ban et al., 2014; Vergara-Temprado et al., 2020; Heim
et al., 2021) albeit without expecting a significant improvement for water vapour isotope variables (de Vries et al., 2022).
Furthermore, we anticipate a depletion bias in the near-surface vapour (Thurnherr et al., 2021) and an enriched bias in the
free-tropospheric vapour but overall a good performance in terms of the mesoscale to synoptic-scale variability in humidity
and isotope variables (Dahinden et al., 2021; de Vries et al., 2022). Motivated by Thurnherr et al. (2021)'s consideration, we
used the smooth regime of Merlivat and Jouzel (1979) in our simulations to formulate non-equilibrium fractionation during
evaporation from the ocean surface.

## 2  Data and methods

### 2.1  Numerical simulations

We carried out three convection-resolving simulations with different grid resolutions using the regional model COSMO$_{iso}$.
The implementation of isotopes in COSMO$_{iso}$ is achieved through a parallel water cycle for each of the two heavy isotopes,
$^1H^2H^{16}O$ and $^1H_2^{18}O$, which differs from the one of $^1H_2^{16}O$ by accounting for fractionation processes. The two additional
water cycles are purely diagnostic and do not affect other model components (Pfahl et al., 2012). Fractionation processes
during soil water evaporation are included through the coupling of COSMO$_{iso}$ and TERRA$_{iso}$, an isotope-enabled prognos-
tic multilayer soil model (for details see Christner et al., 2018). Plant transpiration is treated as a non-fractionating process
(Aemisegger et al., 2015). For evaporation from the ocean surface, the non-equilibrium fractionation factors from the smooth
regime of Merlivat and Jouzel (1979) were used. Cloud processes were simulated by a one-moment microphysics scheme with
a fixed number of cloud droplets ($N_c = 5 \times 10^8$ m$^{-3}$; Doms et al., 2011; Seifert and Beheng, 2001, 2006). The convection
schemes of the model (Tiedtke, 1989; Theunert and Seifert, 2006) were disabled, meaning that deep and shallow convection
was explicitly resolved. A 20 s model time step was used, and hourly output was generated. A nested approach with spectral
nudging of horizontal winds in the free troposphere was chosen to allow for a direct comparison to observations and to correctly
capture influences from large-scale advection via the lateral boundary conditions. Previous isotope-enabled modelling studies
in the tropics mostly adopted a radiative-convective equilibrium setup (Bony et al., 2008; Risi et al., 2008; Blossey et al., 2010)
and produced a negative bias in heavy isotopes of precipitation (Bony et al., 2008; Torri et al., 2017).

The initial and lateral boundary conditions for the COSMO$_{iso}$ simulation with the coarsest spatial resolution, referred to
as COSMO$_{iso, 10km}$, were taken at 6-hourly intervals from a simulation performed with the global model ECHAM6-wiso
(Cauquoin et al., 2019; Cauquoin and Werner, 2021). To reproduce the large-scale meteorological conditions during EUREC$^4$A,
ECHAM6-wiso was nudged (incl. surface pressure, temperature, and horizontal winds) towards ERA5 (Hersbach et al., 2020).
The ECHAM6-wiso simulation was performed with parameterised convection and used the wind-dependent non-equilibrium



fractionation factors from Merlivat and Jouzel (1979) for evaporation from the ocean surface. The global model was run with the spectral grid T127 (approx. $0.9°$ horizontal resolution) and 95 vertical levels.

COSMO$_{iso, 10km}$ has a horizontal resolution of $0.1°$, 40 vertical levels, and covers most of the North Atlantic (Fig. 2, Table 1). Horizontal winds above $850\,hPa$ in COSMO$_{iso,10km}$ were nudged towards ECHAM6-wiso. The spectral nudging technique

ensured that the regional simulations remain close to the large-scale flow in the global model (Von Storch et al., 2000; Schubert-Frisius et al., 2017). The nudging included only zonal and meridional wavenumbers of less than five. COSMO$_{iso,10km}$ covers the period from 6 January to 13 February 2020. The first 10 days are treated as spin up and are not included in the analysis.

The second COSMO$_{iso}$ simulation, COSMO$_{iso,5km}$, has a horizontal resolution of $0.05°$, 60 vertical levels, and covers a subset of the western North Atlantic, including the northern part of the South American continent (Fig. 2, Table 1). Initial and

lateral boundary conditions were taken from COSMO$_{iso,10km}$ at hourly time steps. The spectral nudging technique was identical to the first COSMO$_{iso}$ simulation but nudged towards COSMO$_{iso,10km}$ horizontal winds above $850\,hPa$ instead of ECHAM6-wiso (using the same wavenumbers as for the COSMO$_{iso,10km}$ albeit resulting in smaller wavelengths given the smaller model domain). COSMO$_{iso,5km}$ covers the period from 20 January to 13 February 2020. All simulated days are included in the analysis.

The third COSMO$_{iso}$ simulation, COSMO$_{iso,1km}$, has a horizontal resolution of $0.01°$, 60 vertical levels, and covers the focus

area of EUREC[4]A's field activity (Fig. 2, Table 1). Initial and lateral boundary conditions were taken from COSMO$_{iso,5km}$ at hourly time steps. The spectral nudging, in this case, was directed towards COSMO$_{iso,5km}$ wind data. COSMO$_{iso,1km}$ covers the period from 20 January to 13 February 2020, and all simulated days are taken into account for the analysis.

## 2.2 Observational datasets

### 2.2.1 Satellite observations

The true colour images from the Visible Infrared Imaging Radiometer Suites (VIIRS) onboard the polar-orbiting satellite Suomi-NPP (National Polar-orbiting Partnership; NOAA, 2017) are used to assess the spatial organisation of clouds. The images were retrieved from the Worldview Snapshot application on NASA's Earthdata platform. For the same purpose, the data (with a spatial resolution of $500\,m$) from the visible channel of the satellite GOES-16, which is part of NOAA and NASA's Geostationary Operational Environmental Satellite (GOES) – R Series, were downloaded from the EUREC[4]A data

catalogue.

The Integrated Multi-satellitE Retrievals (IMERG) from the Global Precipitation Measurement (GPM) mission is used for the evaluation of the spatial distribution of precipitation. We refer to the data, which was obtained from NASA's Earthdata platform, simply as GPM. Here, the half-hourly (Huffman et al., 2019a) and daily (Huffman et al., 2019b) precipitation estimates are used.

### 2.2.2 Aircraft observations

Several observational datasets that were created during EUREC[4]A onboard the ATR and the German aircraft HALO (High Altitude and Long Range Research Aircraft; Konow et al., 2021) are used. All of them are available on AERIS (https://eurec4a.



aeris-data.fr, last access: 18 September 2022). The ATR conducted 19 flights on 11 days from 25 January to 13 February
2020. A flight typically lasted four to five hours and consisted of repetitive flight patterns. They included the ferry from the

Grantley Adams International Airport of Barbados into the western half of the so-called EUREC[4]A circle (see definition
below), followed by two or three rectangles (120 km × 20 km) at cloud base, one or two L-patterns flown either near the top or
the middle of the subcloud layer, occasionally a surface leg at an altitude of about 60 m, and the ferry back to the airport, which
often included a short leg above the trade inversion to sample free tropospheric air. The ATR flights were closely coordinated
with those of the HALO, which performed 15 flights from 19 January to 18 February 2020. Each HALO flight lasted about

nine hours, seven of which the aircraft spent at upper altitudes (∼10 km) in the EUREC[4]A circle centred at 13.3° N, 57.7° W
(roughly 150 km east of Barbados), with a diameter of ∼220 km. When the HALO was flying this circle, dropsondes were
launched frequently to estimate the large-scale vertical motions (Bony and Stevens, 2019; Stevens et al., 2021; Konow et al.,
2021; George et al., 2021a, 2022). The HALO dropsonde data (George et al., 2021b) is used for the evaluation of the COSMO$_{\text{iso}}$
simulations.

The ATR's position and the corresponding standard meteorological variables (e.g., pressure, temperature, wind components)
are reported in the CORE dataset (CNRM/TRAMM et al., 2021; Bony et al., 2022, 1 Hz temporal resolution). Considering that
the ATR moved roughly 100 m s$^{-1}$, the spatial resolution of its observations amounts to 100 m for the 1 Hz data and to 1 km
for the 10 s-averaged data.

The ATR-based BASTALIAS dataset (Delanoë et al., 2021) is used for the evaluation of the cloud fraction at cloud base.

It is a data product resulting from the combined observations of the Doppler cloud radar BASTA (Bistatic Radar System for
Atmospheric Studies; Delanoë et al., 2016) and the lightweight backscatter lidar ALiAS (Airborne Lidar for Atmospheric
Studies; Chazette et al., 2020) that were horizontally probing the surroundings of the ATR.

The ATR-based PMA dataset (Microphysics Airborne Platform; Coutris, 2021, 1 Hz temporal resolution) is used for the
microphysical characterisation of the clouds and their surroundings. It contains the combined observations from an optical-

scattering droplet spectrometer (Lance et al., 2010) and an optical array stereo probe imager (Lawson et al., 2006), together
sizing and counting the droplets with diameters from 2 μm to 2 mm. From the size distribution composites of the two instru-
ments, the total concentration of particles, the median volume diameter, the liquid water content (LWC), and flags indicating
the presence of cloud, drizzle or raindrops are inferred. Cloud drops are identified if the LWC exceeds 10 mg m$^{-3}$ and the
particle diameter remains below 100 μm; drizzle if the particle diameter is at least 100 μm but remains below 500 μm; and rain

if the particle diameter exceeds 500 μm.

Finally, the ATR-based Picarro dataset (Aemisegger et al., 2021a) is used for the humidity conditions and the isotopic
composition of the vapour, which were both measured with a customised cavity ring-down spectrometer L2130-*i* from the
manufacturer Picarro with a sampling frequency of 1 Hz (more technical details in Bailey et al., 2023). The Picarro dataset
provides a flag indicating data points of questionable quality (e.g., due to inlet wetting). These data points were removed for

the analyses shown in this paper. Moreover, only 10 s-averages (for a better noise-to-signal ratio) are used, which corresponds
to an effective horizontal resolution of 1 km and thus makes the observations comparable to the COSMO$_{\text{iso,1km}}$ simulation.





Note that the ATR observations have been thoroughly evaluated in Bony et al. (2022). They found that the different ATR measurements of humidity, wind and cloud fraction are in good agreement with each other and with HALO dropsondes.

## 2.3 Definition of relevant atmospheric levels

The ATR data comes with an official flight segmentation (Bony et al., 2022). Here, we use the data that is labelled as subcloud and cloud base. To ensure that only the data points collected at the base of shallow clouds are taken into account, we apply the criterion that only data points with altitudes below 1.3 km (threshold for the distinction between shallow and deep clouds used in Vial et al., 2019) are treated as cloud base observations. The resulting subcloud and cloud base mean/median pressure for each flight are shown in Fig. 3.

Two levels, representing cloud base and the subcloud layer, are defined for the COSMO$_{iso}$ simulations for comparison with the ATR data. Cloud base represents the interface between the subcloud layer and the cloud layer (Fig. 1). The subcloud layer extends from the surface to cloud base (Fig. 1). However, it is assumed to be well-mixed (except for the thin transition layer at its top; Albright et al., 2022) and therefore sampling it at one altitude is sufficient for its characterisation. For simplicity, cloud base and the subcloud layer in COSMO$_{iso}$ are defined by model levels. The two model levels are identified for every hourly

time step individually for the domain 54.5-61° W and 11-16° N. For every time step, we first determine the cloud base level, followed by the identification of the level representing the subcloud layer:

1. Every vertical profile in the domain 54.5-61° W and 11-16° N is checked for cloud water content exceeding 10 mg kg$^{-1}$ (threshold for the detection of clouds used in Vial et al., 2019, also consistent with the threshold used in the above-described PMA dataset) at any model level below 1.3 km. The lowest model level meeting this criterion is taken as the

cloud base of the respective vertical profile. If the profile does not contain any clouds (i.e., does not meet the criteria), it is ignored in the subsequent step.

2. In order to extract cloud base conditions we determine one cloud base model level for the domain 54.5-61° W and 11-16° N by calculating the median over the cloud base model levels identified for cloudy profiles in the previous step.

3. Steps 1 and 2 are repeated for every hourly time step of the simulated time period. The resulting time series of cloud

base model levels can then be used to extract cloud base variables (as, e.g., pressure shown in Fig. 3) from the COSMO$_{iso}$ simulations.

The cloud base levels identified in the three COSMO$_{iso}$ simulations, using the procedure described above, alternate between the model levels 29 (783 m) and 30 (970 m) for COSMO$_{iso,10km}$ and 47 (761 m), 48 (914 m), and 49 (1082 m) for COSMO$_{iso,5km}$ and COSMO$_{iso,1km}$. COSMO$_{iso}$ underestimates the flight-to-flight variability of the cloud base altitude in comparison to the

ATR data (Fig. 3). Nevertheless, the identified COSMO$_{iso}$ cloud base altitudes are in the range of the values observed by the ATR during the cloud base flight segments (Fig. 3b). Therefore, the COSMO$_{iso}$ cloud base data is assumed to be comparable to the ATR cloud base data.

The model level representing the subcloud layer in COSMO$_{iso}$ is identified as the mid-level between cloud base and the surface, which corresponds to the model level 34 (259 m) for COSMO$_{iso,10km}$ and the model levels 53 (233 m) and 54 (309 m)



for $COSMO_{iso,5km}$ and $COSMO_{iso,1km}$. The chosen level within $COSMO_{iso}$'s subcloud layer is lower than flown by the ATR except for two flights (Fig. 3c). Even though we describe the subcloud layer conditions by one single model level, we will refer to it as a layer since the characteristics are expected to be representative of the whole subcloud layer. The subcloud layer is well-mixed with very small vertical gradients in humidity and the isotope variables (confirmed by Fig. 9) such that the ATR and $COSMO_{iso}$ datasets are assumed to be comparable for this layer.

**3   Model evaluation and spatial resolution effects**

EUREC[4]A offers a unique opportunity to perform a thorough evaluation of model simulations targeted at shallow trade-wind cumulus clouds in a process-oriented way thanks to the wealth of available observational data. In the following, variables from the different $COSMO_{iso}$ simulations, which are considered to be relevant for the formation of shallow cumulus clouds in the trade-wind region, are selected and compared to observations. Simultaneously, the effect of the spatial model resolution is

assessed, which is also the reason why the datasets with higher resolutions are not upscaled (e.g., by averaging over space and time) to the one with the lowest resolution. All comparisons are performed with the data in the selected evaluation domain 54.5-61° W and 11-16° N. This domain was chosen large enough to provide a robust statistical basis to analyse cloud properties while remaining small enough to contain a fairly homogeneous cloud field in terms of cloud patterns.

**3.1   Clouds and precipitation**

The $COSMO_{iso}$ simulations are all able to distinguish between different mesoscale organisation patterns (Fig. 4; the daily spatial distribution of clouds over the full simulated period can be found in Supplement 1). Note that the model is not expected to produce clouds identical to the ones in the observations, but it is plausible that the simulations reproduce the mesoscale cloud organisation patterns since they are linked to distinct large-scale environmental conditions (Bony et al., 2020). The elongated Fish clouds on 22 and 26 January 2020 are reproduced by $COSMO_{iso}$, although, at a different location than depicted by the

satellite image, hinting towards slight errors in the propagation speed of the banded clouds associated with trailing cold fronts (see Villiger et al., 2022, for the link between Fish clouds and trailing cold fronts). On 12 February 2020, the satellite shows large Flower clouds, which are not clearly identifiable in the $COSMO_{iso}$ output. Nevertheless, if viewed over a larger domain (Supplement 1), aggregated cloud liquid water content objects are visible, resembling Flower clouds. Smaller Flower clouds, as on 2 February 2020, are clearly reproduced by the model. Finally, finer structured cloud patterns recognised as Gravel and

Sugar, as they appear on 8 and 9 February 2020 in the satellite observations, are also reproduced by $COSMO_{iso}$. Note that the cloud classification used above largely agrees with the patterns identified by Schulz (2022, their Fig. 7a).

    Overall, the spatial cloud patterns from the satellite images are qualitatively better reproduced by the simulations if the horizontal model resolution is increased. This holds especially for the finely structured clouds on 8 and 9 February 2020. For these cases, only $COSMO_{iso,1km}$ shows the Gravel-typical cold pool activity in the form of cloud liquid water arranged in arc-

like structures around clear-sky patches. An exception is 12 February 2020, a day on which the organisation of the clouds in large Flowers is partially lost with the higher resolution of $COSMO_{iso,1km}$.




The four mesoscale organisation patterns discussed above are known to be characterised by different low-level cloud fractions (Bony et al., 2020), peaking at different times of the day (Vial et al., 2021). Therefore, the temporal evolution of cloud fraction at cloud base is another measure to evaluate the performance of the three COSMO$_\mathrm{iso}$ simulations. In all three simula-
tions, the variable is characterised by a pronounced diel cycle with maximum cloud fraction during local early morning (Fig. 5; agreeing with Vial et al., 2019). The magnitude of the cloud fraction, however, is very different for the three simulations. The cloud fraction is largest in COSMO$_\mathrm{iso,10km}$ and becomes progressively smaller with higher resolution until reaching minimum values in COSMO$_\mathrm{iso,1km}$. This behaviour is expected for clouds that are smaller than the grid resolution (explanatory sketch in Supplement 1) and has been observed in earlier studies (Hentgen, 2019; Heim et al., 2021). The comparison to the ATR-based
cloud fraction estimates (symbols in Fig. 5) shows that COSMO$_\mathrm{iso,10km}$ and (for most of the flights) COSMO$_\mathrm{iso,5km}$ overestimate cloud fraction. Contrastingly, COSMO$_\mathrm{iso,1km}$ agrees well with ATR$_\mathrm{BASTALIAS,cloud+drizzle}$ except for the first flight on 2 February and the two flights on 7 February 2020. On these two dates, however, coarse cloud patterns (Flower and Fish; not shown) were present, meaning that the ATR likely underestimated cloud fraction due to its limited sampling area.

In the model, liquid water droplets in a grid cell are separated into the non-sedimenting cloud water content and the sedi-
menting rain water content, respectively. The simulated cloud and rain water contents are summed up for comparison to the ATR-based liquid water content (LWC) measurements provided in the PMA dataset. Consistent with cloud base cloud fraction, the fractions of data points with the $\mathrm{LWC} > 10\,\mathrm{mg\,kg^{-1}}$ are decreasing with increasing spatial model resolution and overall higher in the simulations than in the observations. COSMO$_\mathrm{iso,10km}$ meets the threshold at $22.3\,\%$ of the data points, COSMO$_\mathrm{iso,5km}$ at $14.0\,\%$, COSMO$_\mathrm{iso,1km}$ at $8.3\,\%$, and the PMA at $5.6\,\%$.

The LWC at cloud base is generally overestimated in COSMO$_\mathrm{iso}$ compared to the observations, but the observed values are approached with increasing horizontal model resolution (Fig. 6a). Possibly the number of cloud droplets ($N_c$), which is fixed to $5 \times 10^8\,\mathrm{m^{-3}}$ in the one-moment cloud scheme (Sect. 2), contributes to the offset between simulations and observations. This value is located at the upper end of the observed droplet number distribution (Fig. 6b). A high $N_c$ leads to a reduction of the cloud droplets' size since the available liquid water is distributed over more droplets. This, in turn, reduces the efficiency of
the collisional growth process and can delay or even prevent the formation of precipitation (see, e.g., Eirund et al., 2022). In other words, $N_c$ directly affects the autoconversion rate, the rate at which COSMO$_\mathrm{iso}$ turns cloud water into rain water. Thus, if the parameter were adjusted towards the observations, precipitation would likely be formed more readily (efficiently removing liquid water from the atmosphere), possibly affecting below-cloud evaporation of hydrometeors and surface precipitation. It remains to be tested how COSMO$_\mathrm{iso}$ would react to a change of $N_c$ in the considered regime. We refrain from looking in
more detail into the cloud scheme of COSMO$_\mathrm{iso}$ and leave the comparison of different sensitivity experiments related to the autoconversion rate to future research. The motivation for the considerations above is to draw the attention of the reader to the fact that some model parameters - which can have a substantial influence on the isotope signals - deviate from the observations.

Figure 7 shows the rain water that ultimately reaches the surface. Compared to the satellite product, precipitation falls over smaller areas in COSMO$_\mathrm{iso}$, even if the resolutions of the compared datasets are comparable (COSMO$_\mathrm{iso,10km}$ vs. GPM; keep in
mind that the satellite product is an estimate of surface precipitation and should not be taken as the absolute truth). The patches





with intense rainfall become smaller and more frequent with finer model resolution. Most likely, this is due to an intensification of updrafts with finer model resolution.

Considering that the spatial distribution of precipitation in COSMO$_{iso}$ strongly contrasts with the satellite product (Fig. 7), the rainfall amounts over the evaluation domain are surprisingly close to each other (Fig. 8). It depends on the dominant cloud

pattern, whether COSMO$_{iso}$ is over-/underestimating precipitation and whether the grid resolution has a systematic effect. From 20 to 24 January, when Fish clouds were dominant (Schulz, 2022, their Fig. 7), the simulations show a distinct diel cycle of precipitation (peaking shortly after the peak in cloud fraction; agreeing with Vial et al., 2016), while the satellite reports a period of continuous rainfall (Fig. 8a). Over a 24 h window (Fig. 8b), this leads to a rainfall underestimation on 22 and 24 January 2020 by all three simulations, while on 20, 21, and 23 January 2020 alternately one of the three simulations best matches with

the satellite. The exaggerated diel cycle of precipitation in the simulations highlights potentially too strong radiatively driven mesoscale circulations that may slow down the propagation of the trailing cold front into the tropics (cf., shifted position of the Fish cloud in the simulations compared to the observations; Fig. 4). Note that this period was characterised by anomalous large-scale transport (extratropical dry intrusion discussed in Villiger et al., 2022) with the precipitation-triggering process being dominated by large-scale convergence instead of isolated cells of shallow convection.

From 25 January to 5 February 2020, when the clouds either are classified as Sugar or remain unclassified (Schulz, 2022, their Fig. 7), all COSMO$_{iso}$ simulations overestimate the rainfall amount. The overestimation is smaller the finer the spatial model resolutions. From 6 February 2020 onwards, the rainfall amount is increased in all four datasets, but again none of the three simulations is exceptionally accurate. Much more, it depends on the date, which of the simulations is closest to the observations. This final period of the campaign was again characterised by anomalous large-scale transport (weak extratropical

dry intrusion on 7 and 8 February 2020 and tropical mid-level detrainment starting on 12 February 2020 discussed in Villiger et al., 2022). The dominant cloud patterns in the considered domain were Fish clouds (7 to 8 February 2020) followed by Flower clouds (10 to 12 February 2020; Schulz, 2022, their Fig. 7). Besides 20 to 26 January 2020, the diel cycle of precipitation from COSMO$_{iso}$ agrees with the one from the satellite observations (Fig. 8a).

The variables considered up to this point show a high dependence on the horizontal model resolution. This is not surprising

since the three COSMO$_{iso}$ simulations are in the so-called "grey zone" of resolutions, where only some of the scales involved in convective motions are resolved (Wyngaard, 2004), while the sub-grid-scale processes are taken care of by the turbulence scheme. This interplay between convection and turbulence leads to a strong resolution-dependency of variables associated with convection (Hanley et al., 2015; Jeevanjee, 2017). Whether the increase in resolution leads to a better agreement with observations or not depends on the variable. For instance, the representation of the cloud parameters in the simulations is

improved with finer resolution, while this is not the case for the spatial distribution of precipitation.

## 3.2 Water vapour isotopes, humidity and temperature

In contrast to the variables in the previous section, the variables discussed in this section are largely indifferent to the model resolution (Fig. 9 and 10). Therefore, no distinction between COSMO$_{iso,10km}$, COSMO$_{iso,5km}$, and COSMO$_{iso,1km}$ is made when discussing the three simulations in this section (i.e., COSMO$_{iso}$ is used to refer to all three simulations).



The (close to) saturated layer extends over ∼950-900 hPa in COSMO$_{iso}$ and over ∼950-820 hPa in the measurements from the downward profiles of the ATR, the measurements from the HALO dropsonde measurements, the ERA5 reanalysis, and the ECHAM6-wiso simulation (Fig. 9e). Using high relative humidity (>80 %) as a proxy for the presence of clouds (since observation-based cloud fraction estimates are only available at cloud base), we derive from it that COSMO$_{iso}$ generally has a shallower cloud layer than the other datasets and presumably has difficulties in producing cloud top anvils accurately. A behaviour also observed in other regional model simulations of shallow trade-wind clouds (Beucher et al., 2022, note that they included a shallow convection parametrisation, while we do not). In our COSMO$_{iso}$ simulations, the rather coarse vertical resolution (Sect. 2) might be a reason for the too shallow representation of clouds.

Compared to the measurements from the downward profiles of the ATR, COSMO$_{iso}$ has a clear cold-dry bias of 1.8-2.1° C and 1.6-1.7 g kg$^{-1}$ (Fig. 9a,d and Table 2) in the lower troposphere (from the surface to ∼850 hPa). The HALO dropsonde measurements, ERA5, and ECHAM6-wiso (in which the temperature was nudged) match better with the ATR measurements. ECHAM6-wiso even has a weak moist bias in the cloud layer from about 940 hPa to 700 hPa, meaning that COSMO$_{iso}$ produces its dry-cold bias independently from the ECHAM6-wiso boundary data. Furthermore, COSMO$_{iso}$'s temperature profile shows a slightly deeper adiabatic layer (up to 925 hPa; Fig. 9b,c), which suggests a deeper subcloud layer than in the observations, ERA5, or ECHAM6-wiso (cf., Beucher et al., 2022, who found the same in their simulations). The deeper adiabatic layer leads to a more stable cloud layer, with a stronger increase in virtual potential temperature with height (Fig. 9b) in COSMO$_{iso}$ than in the other datasets. This matches the finding above that COSMO$_{iso}$ has fewer deep clouds (reaching above 850 hPa).

The cold-dry bias in COSMO$_{iso}$ is associated with slightly too depleted vapour compared to the ATR measurements (Fig. 9f,g and Table 2). In contrast, COSMO$_{iso}$ produces too enriched vapour in the free troposphere (above ∼800 hPa). ECHAM6-wiso shows slightly too enriched vapour in the cloud layer and a more accurate decrease in heavy isotopes above cloud tops compared to the ATR observations than COSMO$_{iso}$. The d-excess in vapour (Fig. 9h) is almost constant throughout the vertical column in ECHAM6-wiso and COSMO$_{iso}$. While the values in ECHAM6-wiso overlap with the ATR data (at least up to ∼750 hPa), COSMO$_{iso}$ is shifted towards higher values by ∼4.5 ‰ (Fig. 9h; the bias is visible up to 200 hPa, not shown). Given the fact that ECHAM6-wiso does not have the same isotope biases as COSMO$_{iso}$, it follows that COSMO$_{iso}$ must produce them within the regional model domain. This might be related to the different set of used fractionation coefficients in ECHAM6-wiso vs. COSMO$_{iso}$ (wind-dependent vs. smooth regime of Merlivat and Jouzel, 1979).

The low-level biases in temperature, humidity, and $\delta^2$H are physically consistent. Due to the cold bias alone, a dry and a depletion bias can be expected. In fact, about two-thirds of the bias in $\delta^2$H and one-fourth in the bias of $\delta^{18}$O can be explained by the temperature bias (via the temperature-dependency of equilibrium fractionation). This also means that isotopes can only provide limited information about the origin of the low-level biases. Different mechanisms might possibly cause some of the identified biases at low levels: (1) An overestimation of the evaporation of hydrometeors would explain a low-level cold bias (Fig. 9a). This, however, would simultaneously lead to a moistening, which is not consistent with the dry bias in the COSMO$_{iso}$ profiles (Fig. 9d). (2) A too strong convective mixing (across the full depth of the shown profiles), which transports dry-depleted air downward and moist-enriched air upwards, would fit the observed low-level dry-depletion bias, but only partly the conditions in the free troposphere with the too enriched vapour but no moist bias (Fig. 9d,f,g). Moreover, such a mixing



process is expected to lead to a low-level warm bias (downward transport of high potential temperature and upward transport of low potential temperature) and not a low-level cold bias as observed here (Fig. 9a). Lastly, the underestimation of clouds reaching above 900 hPa (Fig. 9e) does not directly support too strong mixing. If convective mixing was too strong, the few higher-reaching clouds in COSMO$_{iso}$ would need to mix much more efficiently than their more abundant shallow counterparts in the other datasets. (3) Another possible explanation for the cold bias that has not been investigated here but has been found in earlier COSMO evaluation studies (Heim et al., 2021) is too high outgoing longwave radiation (top-of-the-atmosphere) due to a too dry free troposphere or too few low-level clouds. However, we don't observe a dry bias in the free troposphere (Fig. 9d). Furthermore, COSMO$_{iso}$'s cloud base cloud fractions are rather too high compared to the ATR-based estimates (Fig. 5). (4) Finally, we cannot rule out that the biases are caused by the spectral nudging (horizontal winds above 850 hPa) applied for the simulations (e.g., Wehrli et al., 2018; Sun et al., 2019). However, in another study (Beucher et al., 2022), a cold-dry bias was also found in free-running simulations that were performed without spectral nudging.

Two mechanisms might explain the enrichment bias above 800 hPa (Fig. 9f,g): (1) Too small precipitation efficiencies (i.e., too low autoconversion rate) could explain an enrichment bias of about 10‰ (see Rayleigh model including precipitation efficiency in Noone, 2012). However, the magnitude of the bias observed here (offset of roughly 60‰ for $\delta^2H$) is too large to solely be explained by this effect. (2) Another possible explanation is that too strong cloud top evaporation (promoting too many, too small cloud droplets; see discussion regarding cloud droplet number in Sect. 3.1) driven by too strong turbulent mixing at the trade inversion, could lead to only weak moistening but strong enrichment due to total (thus non-fractionating) droplet evaporation. The large isotope bias collocated with a very small or inexistent humidity bias in the region of the trade inversion and the level of the deepest cloud tops illustrates that the isotopes contain additional, complementary information for assessing some model biases

Possible reasons for the d-excess bias (Fig. 9h) are the use of inappropriate non-equilibrium fractionation factors, wrong sea surface temperatures, or too strong near-surface humidity gradients. (1) For the non-equilibrium fractionation factor, evidence is available that the chosen value in COSMO$_{iso}$ can still be improved compared to observations (Supplement 2 and Zannoni et al., 2022). The wind-dependent formulation of Merlivat and Jouzel (1979), used in ECHAM6-wiso, and the output from a sensitivity experiment with COSMO$_{iso}$ with a new wind-dependent formulation of the non-equilibrium fractionation factor from work done by Dütsch et al. (2023, see experimental simulation shown in Supplement 4) yield d-excess values closer to the ATR observations than the evaluated COSMO$_{iso}$ simulations. (2) Sea surface temperatures in COSMO$_{iso}$ are identical to the values from ECHAM6-wiso. Therefore, these alone should not be the origin of the bias. (3) However, in combination with the low-level dry bias in COSMO$_{iso}$, they result in stronger near-surface humidity gradients, which certainly contribute to the high bias in d-excess.

The slightly deeper subcloud layer (adiabatically stratified; Fig. 9b,c) of COSMO$_{iso}$ in comparison to the ATR and HALO observations, as well as ECHAM6-wiso and ERA5 data is likely due to too strong turbulent mixing in the subcloud layer and at cloud base. Another reason could be the differences in the vertical resolution of the compared datasets. Indeed, ERA5 has roughly twice as many model levels in the layer 1000 to 850 hPa as the three COSMO$_{iso}$ simulations. However, ECHAM6-





wiso has fewer levels in the considered layer. Thus, it is not plausible that the deeper adiabatic layer in COSMO$_{iso}$ originates
from a vertical resolution effect.

COSMO$_{iso}$'s low-level biases are further investigated by focusing on cloud base and subcloud values, the two levels at
which the ATR conducted most observations. In line with the vertical profiles, COSMO$_{iso}$ produces cloud base conditions that
are systematically too cold (Fig. 10a), too dry (Fig. 10b), and too depleted (Fig. 10c), together with too high d-excess values
(Fig. 10d). Except for the temperature, similar biases are found in the subcloud layer. The subcloud temperature in COSMO$_{iso}$
remains almost constant over all flights and is sometimes above and sometimes below the ATR observations (Fig. 10a), but
overall also too cold (Fig. 9a).

Despite these offsets, the correlations in Table 3 suggest that the temporal evolution from flight to flight observed by the
ATR is mostly captured by COSMO$_{iso}$. For specific humidity, the temporal correlations between the simulations and the ob-
servations are high in the subcloud layer (0.65-0.73) and low at cloud base (0.25-0.35). For $\delta^2$H, it is the other way around,
with high correlations at cloud base (0.6-0.69) and lower correlations in the subcloud layer (0.45-0.5). The d-excess shows
high temporal correlations at both levels (0.66-0.74). For temperature, correlations between the two datasets are low at cloud
base and negligible in the subcloud layer due to the previously mentioned small variability simulated by COSMO$_{iso}$. The high
correlation for d-excess at both levels is consistent with the fact that the d-excess is sensitive to large-scale drivers, as shown by
previous studies (Pfahl and Wernli, 2008; Aemisegger et al., 2021b), which are relevant for both levels. Not only the temporal
correlation but also the small in-flight variability of the d-excess support this hypothesis.

The smaller vertical differences between cloud base and subcloud values of humidity, $\delta^2$H, and d-excess in COSMO$_{iso}$
compared to the ATR (Table 4) are in line with the above-stated hypothesis of a too strong turbulent mixing at low levels
in the simulations. Associated with locally too strong turbulent mixing, the deeper adiabatic layer (Fig. 9b,c) leads to larger
differences between subcloud and cloud base temperatures in COSMO$_{iso}$ than in the ATR data (Table 4). The in-flight vari-
505 ability of the subcloud temperatures (Fig. 10a) is underestimated by COSMO$_{iso}$. This is an additional indication of too strong
low-level mixing processes, which too rapidly destroy gradients that are created by thermals or dry-warm tongues. Specific hu-
midity (Fig. 10 b) and $\delta^2$H (Fig. 10 c) generally show a larger in-flight variability at cloud base compared to the flight-to-flight
variability, in the simulations and the observations. This cloud base variability is the topic of the following Sect. 4.

## 4  Drivers of the horizontal variability of isotopes at cloud base

The detailed examination of the observed and simulated isotope signals at cloud base in the previous section (Fig. 10) showed
that $\delta^2$H has a large spatial (in-flight) and temporal (flight-to-flight) variability, while the d-excess varies mainly from flight to
flight. The flight-to-flight variability is related to the variability of the synoptic circulation. From earlier studies, it is known
that the d-excess is negatively correlated with the distance to the moisture source (Aemisegger et al., 2021b), which is closely
linked to the prevailing large-scale circulation. However, the drivers behind the mesoscale (in-flight) variability of isotopes at
515 cloud base have not been investigated so far. Here, this topic is addressed by looking at the isotopic characteristics of different
environments at cloud base. First, features at the cloud base level are defined in the observational and simulated data based on



a case study on 2 February 2020. Second, the whole datasets are used for a statistical description of these features. Based on the findings of Sect. 3, only the data from the COSMO$_{iso,1km}$ simulation is considered here. Compared to the lower-resolution COSMO$_{iso}$ simulations, COSMO$_{iso,1km}$ reproduces the spatial cloud patterns, cloud base cloud fraction and liquid water content better.

## 4.1 Detailed case study

On 2 February 2020, the ATR conducted research flight 9 (RF09) and 10 (RF10) (Bony et al., 2022). Here, we focus on RF10 for an illustrative case study. RF10 included three rectangles at cloud base. During these, the aircraft sampled clear-sky environments at both edges of the rectangle and a Flower cloud in-between (Fig. 11a,b and 12e). Beneath the Flower cloud, a cold pool is spreading radially away from the cloud centre (Fig. 11a,b). The edges of the cold pool are clearly marked by shallow clouds arranged in an arc at the edge of a clear-sky patch visible in the satellite image (blue dashed lines in Fig. 11a,b). The ATR cloud base rectangle is positioned such that its southern tip lies inside (or above) the cold pool, while its northern tip lies outside.

The four sondes dropped from the HALO in close proximity to the flight track of the ATR demonstrate how different the environments in and around the Flower cloud were (note that the snapshots of the satellite images in Fig. 11 do not correspond to the exact launch times of the dropsondes, which require approximately $10\,\mathrm{min}$ to reach the surface assuming a launch altitude of $10\,\mathrm{km}$ and an average fall speed of $16\,\mathrm{m\,s^{-1}}$ based on the value range suggested in George et al., 2021a). Looking at the measured profiles from the perspective of the ATR flying at cloud base ($\sim$940 hPa; sandy line in Fig. 12e), one can distinguish between a profile with dry-warm conditions at cloud base (red profile in Fig. 11), two profiles with humid-cold conditions at cloud base (yellow and green profiles in Fig. 11), and a profile with a deep cloud layer aloft (blue profile in Fig. 11).

The blue profile with the deep cloud layer aloft, near the centre of the Flower cloud, bears the imprint of the cold pool in the subcloud layer. It is characterized by a saturated layer from 900 to 700 hPa (Fig. 11c), which most likely led to the formation of rain (also observed by the ATR, see Fig. 12e). Below the cloud layer, isothermal conditions prevail (Fig. 11e) while relative humidity (Fig. 11c-d) rapidly decreases (promoting the evaporation of falling hydrometeors). At about 975 hPa, relative humidity reaches a local minimum value (Fig. 11c), which is co-located with a temperature drop of about 2 K (Fig. 11e), pointing towards a strong evaporatively triggered cold pool. A similar, although weaker, temperature drop is visible in the green profile, located at the edge of the cold pool. The comparably shallow mixed-layer height in the two (blue and green) soundings matches the definition of cold pool soundings applied in Touzé-Peiffer et al. (2022).

Although the yellow profile is located inside the cold pool (Fig. 11a,b), it doesn't show the cold-pool characteristic temperature drop near the surface. Possibly, the differences between the green and yellow profiles are related to their position relative to the cold pool centre. Towards the east, where the yellow profile is located, the radially spreading cold pool moves against the general flow of the easterlies, seemingly leading to a quick alteration of the air's cold-pool characteristics.

Lastly, as expected from its position outside the cold pool (Fig. 11a,b), the red profile is totally unaffected by the cold air spreading near the surface. Instead, it shows an adiabatic profile, typical for the well-mixed subcloud layer.



At the altitude flown by the ATR, the largest horizontal contrasts in humidity and temperature are found between the cloudy (green) and warm clear-sky (red) dropsonde profiles, which match with the observations of the ATR (Fig. 12). In the clear-sky background environments at the northern edge of the rectangle (at $13.8°$ N; Fig. 12f), specific humidity and $\delta^2$H in vapour reach minimum values (Fig. 12b), while a level-relative maximum is reached in temperature (Fig. 12c). On the long side of the rectangle, inside cloudy flight sections (Fig. 12e and grey shadings in all panels), specific humidity and $\delta^2$H in vapour

are higher, and temperature is lower compared to their respective mean values. Regarding the turbulence intensity indicated by the vertical velocity fluctuations (relative to the flight-segment mean; Fig. 12d), the cloudy sections are quite turbulent with amplitudes of fluctuations up to $2\,\mathrm{m\,s^{-1}}$, while the clear-sky background environments exhibit weak turbulence close to laminar flow conditions. For the d-excess (Fig. 12a), it is not possible to make a statement about a possible influence of the immediate surroundings (cloud vs. clear-sky) due to the noise of the measurements.

The fact that the clear-sky environment north of the Flower cloud distinguishes itself from the one in the south by lower humidity and higher potential temperature leads to the conclusion that it is a region where air masses subside (see the location of the ATR icon in the schematic Fig. 1a), balancing the upward air mass transport inside the clouds. The clear-sky environment at the southern edge of the cloud base rectangle has no particular anomalies in any observed variables. Possibly the characteristics of this southern clear-sky environment are shaped by dissipated former clouds (i.e., high humidity; cf., Albright et al., 2022).

Coming back to the original purpose of this section (i.e., to study the in-flight isotope variability at cloud base), the cloud base time series of the ATR (Fig. 12) clearly show that the variability in $\delta^2$H in the vapour is driven by the contrast between cloudy and dry-warm, clear-sky background environments. Thus, the question arises whether these patterns are reproduced by the COSMO$_{\mathrm{iso}}$ simulations.

On 2 February 2020 at 15 UTC, a large, partially precipitating cloud, resembling a Flower cloud, appears in COSMO$_{\mathrm{iso,1km}}$

just north of the ATR flight track (Fig. 13). The humid environment of the cloud is flanked to the west and east by dry-warm patches (Fig. 13a,b and 14a). These dry-warm blobs are characterised by low $\delta^2$H (Fig. 13c and 14b) and low d-excess (Fig. 13d and 14c) in vapour. Thus, COSMO$_{\mathrm{iso,1km}}$ produces dry-warm features at cloud base, which match the above description based on the ATR and HALO dropsonde observations. Furthermore, it is the contrast between the dry-warm blob and the cloudy environments (Fig. 13c) that shape the width of the distribution of simulated $\delta^2$H values, similar to what has been observed in

the ATR cloud base measurements.

Putting these findings into the context of the theory summarised in the introduction (Sect. 1), it follows that the variability of $\delta^2$H at cloud base mirrors the mesoscale transport as cloudy environments are more enriched (updraft of freshly evaporated moisture) compared to dry-warm environments (subsidence of depleted vapour from aloft). This, however, does not mean that microphysical processes do not influence $\delta^2$H as well. Depletion of the vapour inside the cloud due to the formation of liquid

droplets most likely reduces the observed horizontal contrasts between cloudy and dry-warm environments at cloud base. The imprint of these local processes on $\delta^2$H in the vapour is, however, much smaller than the signal of the mesoscale transport that reflects the cloud processing integrated over the mesoscale moisture cycle.



## 4.2 Statistical evaluation

To assess the robustness of the insights gained from the 2 February 2020 case, objective criteria to detect the different cloud
base environments in the two datasets are introduced. For the ATR, the flags from the PMA dataset (Bony et al., 2022) are used
to distinguish between time steps in clear-sky conditions and time steps with liquid water droplets in the surroundings. The
following categories are defined (see the percentage of data points in each category in Table 5):

- *clear*: time steps with no PMA flags,

- *cloud*: time steps with the PMA flags indicating clouds, but neither rain nor drizzle,

- *cloud-rain*: time steps with the PMA flags indicating clouds and rain.

Note that the PMA flags indicate rain but no clouds for $1.3\%$ of the time (compared to $0.2\%$ indicating rain and clouds).
These time steps are excluded from the categorisation because they complicate the interpretation of the isotope signal (highly
depending on the formation altitude of the rain). The time steps identified as *clear* are relabelled as *dry-warm* if they have
a negative anomaly in specific humidity ($q$) and a positive anomaly in potential temperature ($\theta$). The anomalies are defined
relative to each flight's mean $\overline{X}$ and standard deviation $\sigma(X)$ of the respective variable $X$ at cloud base with the criteria
$q < \overline{q} - \sigma(q)$ and $\theta > \overline{\theta} + \sigma(\theta)$ (e.g., red lines in Fig. 12b,c), which must be fulfilled for a time step to be assigned to the *dry-warm* category. The two categories *clear* and *dry-warm* are exclusive (i.e., data points identified as *dry-warm* are removed from
the *clear* category).

For the COSMO$_{\text{iso,1km}}$ data, the cloud base grid points in the evaluation domain 54.5-61° W and 11-16° N are stratified
according to their cloud and rain water contents:

- *clear*: grid points with cloud water content below $10\,\text{mg}\,\text{kg}^{-1}$ and rain water content below $1\,\text{mg}\,\text{kg}^{-1}$,

- *cloud*: grid points with cloud water content exceeding $10\,\text{mg}\,\text{kg}^{-1}$ (black contours in Fig. 13) and rain water content
  below $1\,\text{mg}\,\text{kg}^{-1}$,

- *cloud-rain*: grid points with cloud water content exceeding $10\,\text{mg}\,\text{kg}^{-1}$ (black contours in Fig. 13) and rain water content
exceeding $1\,\text{mg}\,\text{kg}^{-1}$ (green contours in Fig. 13).

From the *clear* category, the grid points with a negative anomaly in $q$ and a positive anomaly in $\theta$ are reassigned to the *dry-warm* category, similarly as it is done for the measurements. Here, the anomalies are defined grid-point-wise, by removing the
daily cycle. The hour-of-the-day mean and standard deviation for each grid point are calculated over the whole simulated period
(20 January to 13 February 2020). Note that for the ATR observations, the effect of the daily cycle could not be subtracted
because too few observations are available, in particular at night, preventing a robust estimation of the daily cycle. *Dry-warm*
grid points are then select using the following criteria: $q_{i,t} < \overline{q_{i,h(t)}} - \sigma(q_{i,h(t)})$ and $\theta_{i,t} > \overline{\theta_{i,h(t)}} + \sigma(\theta_{i,h(t)})$ ($i$ indicates the
grid points in the evaluation domain, $t$ the hourly time steps of the simulated period, and $h(t)$ the hour of the day corresponding
to the time step $t$). An overview of the number of identified *dry-warm* grid points per time step is given in Supplement 3 and



for the case study on 2 February 2020 the *dry-warm* grid points are shown as red dots in Fig. 13a. The grid points with rain water content exceeding $1\,\mathrm{mg\,kg^{-1}}$ but remaining below the cloud water content threshold of $10\,\mathrm{mg\,kg^{-1}}$ are not assigned to any category. Such grid points make up $0.6\,\%$ of the cloud base grid points over the hourly time steps closest to the ATR flights (compared to $0.3\,\%$ with rain and clouds).

Applying the above-described categorisation to the full ATR and COSMO$_{\mathrm{iso,1km}}$ datasets yields similar gradients in the cloud base environments (Fig. 15) as observed on 2 February 2020 (Fig. 12, 13 and 14). Both datasets describe a gradient from dry-depleted to moist-enriched conditions going from *dry-warm* to *cloud/cloud-rain* environments (Fig. 15a,b). The specific humidity anomalies of the different cloud base environments are comparable in the ATR and the COSMO$_{\mathrm{iso}}$ data (Table 6). The $\delta^2\mathrm{H}$ anomalies are comparable for the *dry-warm* category. The *cloud* and *cloud-rain* environments, however, are more enriched relative to the level mean in the ATR data than in the COSMO$_{\mathrm{iso}}$ data.

Due to the chosen statistical approach, we can now also identify contrasts in the d-excess at cloud base with negative anomalies in the *dry-warm* blobs and positive anomalies inside clouds (Fig. 15c). However, the d-excess distributions of the different cloud base environments overlap a lot, both in the ATR data and the simulations (especially if the full width of the distribution is considered as in Supplement 3). The direction of the observed d-excess gradients does not fit the general expectations of high d-excess being associated with low relative humidity and vice-versa (Bony et al., 2008). Looking at the vertical sections on 2 February 2020 (Fig. 14c), one notes that d-excess is high near the surface, where ocean water is evaporated under non-equilibrium conditions (Aemisegger and Sjolte, 2018), and slightly decreases with altitude due to shallow cloud processing (Thurnherr et al., 2021). Inside the lower part of the clouds, however, d-excess remains high, especially in the vicinity of strong updrafts (Fig. 14d). Additionally, as d-excess is higher below precipitating clouds (Fig. 14c at about -60.5° E and -58.2° E) than below non-precipitating clouds (Fig. 14c at about -55.9° E), partial evaporation of falling hydrometeors may additionally increase the d-excess in the subcloud layer vapour, which is then transported to cloud base in convective updrafts. In other words, the observed contrasts in d-excess between *cloudy* and *dry-warm* environments at cloud base, similar to $q$ and $\delta^2\mathrm{H}$, reflect mesoscale transport processes.

The full distribution width of cloud base anomalies (black in Fig. 15) is explained by the contrast between the different cloud base environments. In the case of specific humidity and $\delta^2\mathrm{H}$, the *dry-warm* and *cloud-rain* categories are centred (median value) outside of the interquartile range of the full distribution, while it is the other way around for the d-excess. This applies to both datasets, the observations and the simulations. It means that the d-excess is not as strongly influenced by the mesoscale circulation associated with clouds as $\delta^2\mathrm{H}$ and specific humidity. Other factors (e.g., spatial gradients as in Fig. 13d) play an equally or even larger role in the formation of d-excess anomalies.

We have a look at the mean vertical velocity (fluctuations; in case of the ATR data) inside the different cloud base environments to support the assumption that the *dry-warm* and *cloud/cloud-rain* categories represent different branches of the mesoscale overturning circulation (Fig. 15d). In the COSMO$_{\mathrm{iso,1km}}$ data, the mean vertical velocity is negative in the *dry-warm* and *clear* environments and positive in the *cloud* and *cloud-rain* environments as we would expect from an overturning circulation. Similar contrasts between the cloud base environments are found in the ATR data for the mean vertical velocity fluctuations. Note, that for the ATR data, we look at fluctuations relative to the flight-segment mean, because the accuracy of



0.1 m s⁻¹ of the aircraft's vertical motion measurements is not sufficient for estimating mean vertical winds (Lenschow et al.,
650   2007).

## 5   Summary and conclusion

A combination of observational data from EUREC⁴A was used to evaluate COSMO$_{iso}$ simulations at different horizontal reso-
lutions with a focus on processes at cloud base, which is the rooting level for cloud formation. This is the first isotope-enabled
modelling study of shallow cumulus clouds in the trades with a storm-resolving model and in real large-scale meteorological
conditions (i.e., applying a spectral nudging of the free-tropospheric horizontal winds). The detailed process-based evaluation
focusing on shallow clouds was possible thanks to the wealth of observations collected during EUREC⁴A, in particular on
board the French ATR-42 aircraft.

The comparison to satellite images showed that COSMO$_{iso}$ is capable to distinguish between Fish clouds, Flower clouds,
and finer structured cloud patterns, such as Gravel and Sugar. Overall, if the model resolution is high, the simulated mesoscale
cloud organisation agrees better with the satellite images than for simulations with lower resolution. The model differentiates
between periods with high-intensity (20-24 January and 6-13 February 2020) and low-intensity rainfall (period in between),
which are evident in satellite-based observations. For the major part of the simulated period, the model output matches the
diurnal cycle of the satellite-based precipitation product and earlier studies (Vial et al., 2019). The flight-to-flight, as well as
the in-flight variability of the humidity and isotope signals at cloud base and in the subcloud layer, are well captured by the
model (similar to earlier studies looking at the meso- to synoptic-scale variability in humidity and isotope variables; Dahinden
et al., 2021; de Vries et al., 2022).

However, for some variables, COSMO$_{iso}$ has important biases. The liquid water content and cloud fraction at cloud base are
overestimated but approach the observed values the higher the spatial model resolution (for cloud fraction, this behaviour is in
agreement with earlier studies, i.e., Heim et al., 2021). Precipitation is spatially more concentrated in the simulations than in the
satellite product. This local concentration of precipitation increases with higher model resolution. We find biases in humidity,
temperature, and water vapour isotopes independently of the spatial model resolution. COSMO$_{iso}$ produces a distinct cold-dry
bias at low levels that is consistently associated with too depleted vapour. A cold-dry bias was previously observed in other
regional models such as HARMONIE (Van der Voort, 2021, humidity bias of similar magnitude; smaller bias in temperature)
and AROME (Beucher et al., 2022, both biases of smaller magnitude), while too depleted vapour near the surface was found
in COSMO$_{iso}$ simulations performed by Thurnherr et al. (2021, bias of approx. 2-4‰) over the Southern Ocean. Furthermore,
COSMO$_{iso}$ shows too enriched vapour above the inversion (in agreement with Dahinden et al., 2021; de Vries et al., 2022) but
no moist bias. This shows the additional value of isotope observations compared to specific humidity alone and hints towards
a misrepresentation of cloud top processes and their interaction with the dry free troposphere in the model. The secondary
isotope variable d-excess is too high from the surface up to 200 hPa. Too high d-excess values were also found by Thurnherr
et al. (2021). However, here, we used a different formulation of the non-equilibrium fractionation factors, which makes a direct
comparison difficult. Different mechanisms that may cause individual biases were discussed, but no general explanation was



found for their combined occurrence. It seems likely that COSMO$_{iso}$'s turbulent mixing in the subcloud layer up to cloud base is too strong. This would explain the underestimation of the vertical gradients (humidity, $\delta^2$H, and d-excess) from the subcloud to the cloud base level as well as the underestimation of the subcloud temperature variability in comparison to observations.

We recommend a number of additional analyses and sensitivity studies for future research. The role of turbulent mixing should be tested by adjusting the mixing length of the turbulence scheme. Mixing processes in the subcloud layer could be further investigated by combining turbulence and isotope observations from the ATR or by investigating the three-dimensional fields of total kinetic energy in the COSMO$_{iso,1km}$ data. Furthermore, subcloud layer features such as dry tongues and thermals could be assessed in more detail. The role of the fixed cloud droplet number, which is large compared to the observed spectrum,

for the spatial distribution of cloud and rainwater should be assessed. Cloud top processes in the COSMO$_{iso}$ data could be investigated in more detail since near the trade inversion isotopes might provide interesting additional information compared to specific humidity alone (formation of the stratiform outflow and sublimation of cloud droplets). Furthermore, we suggest repeating the simulations without applying spectral nudging to assess its potential role in the formation of the low-level cold-dry and depletion bias. In addition, a better fitting formulation of the non-equilibrium fractionation factors should be found to

reduce the bias in the d-excess (cf., Supplement 4).

Based on the evaluation presented here, we recommend using the highest resolution simulation COSMO$_{iso,1km}$ to study cloud processes in the lower troposphere of the trade-wind region. Compared to COSMO$_{iso,10km}$ and COSMO$_{iso,5km}$, this simulation yields the most realistic spatial distributions of clouds, cloud fraction, liquid water content, and daily amount of precipitation. Nevertheless, it must be kept in mind that COSMO$_{iso,1km}$ was no better in terms of model-internal biases in humidity, temper-

ature, and isotope signals in vapour. Even if COSMO$_{iso,1km}$ performs best in many aspects, the coarse-resolved simulations are still useful for looking at the role of the larger-scale circulation as a driver for shallow cloud organisation.

Following these findings, COSMO$_{iso,1km}$ was used for a feature-based evaluation (distinguishing between different environments named *dry-warm*, *clear*, *cloud*, and *cloud-rain*) investigating the isotope variability at cloud base. During individual flights, $\delta^2$H was found to be mainly controlled by mesoscale transport (overturning circulation associated with clouds) and

not by local microphysical processes. The heaviest vapour with the highest d-excess was found inside clouds (directly linked to shallow convective upward transport), and the lightest vapour with the lowest d-excess inside dry-warm, clear-sky environments (directly linked to mesoscale subsidence), in the observations as well as in the COSMO$_{iso,1km}$ data. In addition, the feature-based evaluation showed that the low-level biases in the COSMO$_{iso}$ simulations discussed above are not feature-specific but apply to all of the different cloud base environments (cf. Supplement 3). In other words, the processes linked to the for-

mation of these features are accurately reproduced by COSMO$_{iso}$. The clear relation between isotope signals and cloud base features reflecting the mesoscale circulation raises the question whether isotope signals at cloud base contain information on the strength of the cloud-relative overturning circulation. This question is addressed in the second part of this study.

*Data availability.* The ERA5 reanalysis dataset (Hersbach et al., 2020) can be downloaded from the ECMWF's official website (https://www.ecmwf.int/en/forecasts/datasets/reanalysis-datasets/era5, last access: 29 October 2022). The daily (Huffman et al., 2019b, https://doi.



org/10.5067/GPM/IMERGDF/DAY/06, last access: 29 October 2022) and half-hourly (Huffman et al., 2019a, https://doi.org/10.5067/GPM/ IMERG/3B-HH/06, last access: 29 October 2022) GPM IMERG data is available on NASA's Earthdata platform (https://disc.gsfc.nasa.gov/, last access: 6 April 2022). The six Suomi satellite images of the Earth Observing System Data and Information System (EOSDIS) are available in the Worldview Snapshots application (https://go.nasa.gov/3SOFwJw; https://go.nasa.gov/3fDXPCP; https://go.nasa.gov/3rsf1Oq; https://go.nasa.gov/3RvSxGA; https://go.nasa.gov/3e43ZvB; https://go.nasa.gov/3Cs4au5; last access: 3 October 2022). The aircraft-based

observations, i.e., the CORE (CNRM/TRAMM et al., 2021), the BASTALIAS (Delanoë et al., 2021), the PMA (Coutris, 2021), the Picarro (Aemisegger et al., 2021a), and the JOANNE (George et al., 2021b) data can be obtained from the EUREC$^4$A data repository AERIS (https://eurec4a.aeris-data.fr, last access: 18 September 2022). The data from the GOES-16 visible channel (no. 2) can be downloaded from the EUREC$^4$A data catalogue (https://observations.ipsl.fr/thredds/catalog/EUREC4A/SATELLITES/GOES-E/catalog.html, last access: 18 September 2022). The ECHAM6-wiso are available upon request (contact: Martin Werner), and the COSMO$_{iso}$ simulations are published

in the ETH research collection (Villiger and Aemisegger, 2022, note that the simulations have been extended and now provide data for the period from 20 January to 20 February 2020).

*Author contributions.* FA designed the project and acquired the funding. LV, with the support of FA, carried out the COSMO$_{iso}$ simulations, performed the data analysis, and wrote the paper, on which all the co-authors commented. LV, FA, and HW discussed the results and the structure of the paper in detail. SB initiated and coordinated the ATR observations. LV, MD, SB, ML, PEB, PC, PCo, JD, CF, AS, and

FA carried out the observations onboard the ATR during EUREC$^4$A and assisted in their interpretation. FA, MD, SP were involved in the development and the technical setup of the COSMO$_{iso}$ simulations and assisted in their interpretation. LV, FA, ML, PEB discussed various possible definitions of warm-dry blobs at cloud base and their characteristic signature in the ATR observations. MW performed the ECHAM6-wiso simulation.

*Competing interests.* The authors have no competing interests to declare.

*Acknowledgements.* We thank Urs Beyerle (ETH Zürich) for the technical support regarding the cluster on which the model calculations were performed and Fabienne Dahinden for the help with the installation of the model on the cluster. The ATR aircraft was operated by SAFIRE, the French facility for airborne research, an infrastructure of the French National Center for Scientific Research (CNRS), Météo-France and the French National Center for Space Studies (CNES). We are grateful to SAFIRE for the realisation of the EUREC$^4$A airborne operations. We acknowledge the MeteoSwiss and the ECMWF for the access to the ERA5 reanalyses, NASA for the access to the satellite-based data,

and all those who were involved in the EUREC$^4$A observations used in this publication.

*Financial support.* LV received funding from the Swiss National Science Foundation (grant no. 188731). EUREC$^4$A was funded by multiple national and international organisations, including the European Research Council (ERC; grant no. 694768), the Max Planck Society (MPG), the German Research Foundation (DFG), the German Meteorological Weather Service (DWD), the German Aerospace Center (DLR), and



the National Centre for Space Studies CNES (EECLAT proposal). The cloud in situ data was collected using instruments from the French

Airborne measurement Platform, a facility partially funded by CNRS/INSU and CNES. SP acknowledges funding from the German Research

Foundation (DFG, Deutsche Forschungsgemeinschaft), project ID 441025101.



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

## Appendix A: Description of supplement material

This paper is supplemented by the following four documents:

**Supplement 1** contains an explanatory sketch of the cloud fraction's resolution dependency and further illustrates the spatial
distribution of clouds as in Fig. 4, but for every day from 20 January to 13 February 2020. With this, we want to create
transparency in how well the mesoscale cloud organisation patterns are reproduced by the simulations beyond the six time
steps shown in the paper. Additionally, we include a version of the figures that displays a larger geographical domain than
shown in Fig. 4 with the goal of informing about the cloud organisation on the large scale.

**Supplement 2** provides further evaluation analyses addressing the horizontal wind components and the mesoscale vertical
velocities at cloud base. The analysis should give an idea of how much the dynamical variables are constrained by the spectral
nudging.

**Supplement 3** informs about the number of identified *dry-warm* cloud base grid points in COSMO$_{iso,1km}$. Additionally, it
contains further characterisations of the different cloud base environments (alternative version of Fig. 15). First, the environ-



ments' anomaly distributions are shown in their full widths (instead of the 25-75-percentile and 10-90-percentile range as in Fig. 15) to illustrate their mutual overlap. Second, the absolute values (instead of anomalous values as in Fig. 15) are shown to demonstrate that all cloud base environments in the simulations are characterised by a bias compared to the ATR observations.

**Supplement 4** compares $COSMO_{iso}$ simulations with different formulations of the non-equilibrium fractionation factors. Namely, the $COSMO_{iso,5km}$ and $COSMO_{iso,1km}$ simulations described in the paper (using the smooth regime of Merlivat and Jouzel, 1979) and two $COSMO_{iso}$ simulations, which use the recently developed formulation by Dütsch et al. (2023). The analysis shows that the choice of the non-equilibrium fraction factors clearly influences the simulated d-excess in vapour.





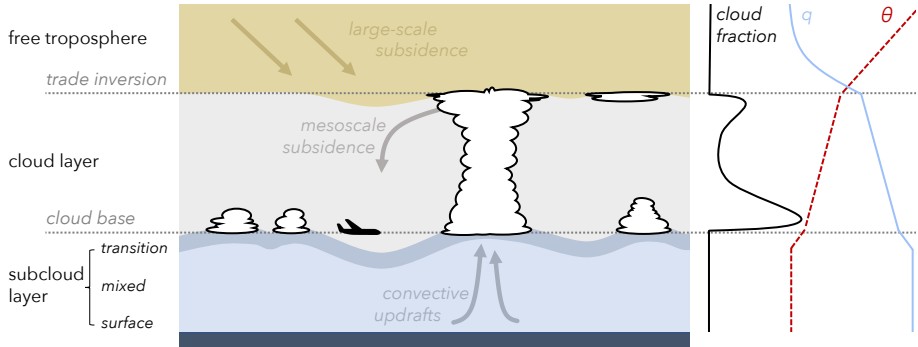

**Figure 1.** Schematic of the different atmospheric layers in the troposphere over the tropical Atlantic with the ATR aircraft flying at cloud base together with idealised profiles of cloud fraction (black), specific humidity (blue) and potential temperature (red dashed). Based on the ATR measurement time series shown in Fig. 12, we assume that the cloud base/subcloud layer top is characterised by an uneven topography instead of being a flat surface. See the text for the description of the individual layers.



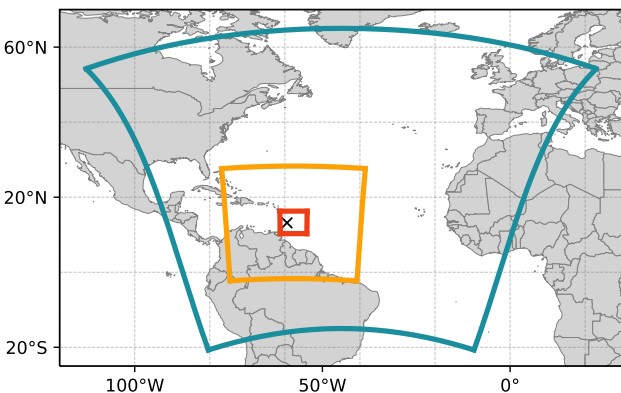

**Figure 2.** Domain boundaries of the three COSMO$_{iso}$ simulations, COSMO$_{iso,10km}$ (teal), COSMO$_{iso,5km}$ (yellow), and COSMO$_{iso,1km}$ (red). The location of Barbados is indicated by a black cross.





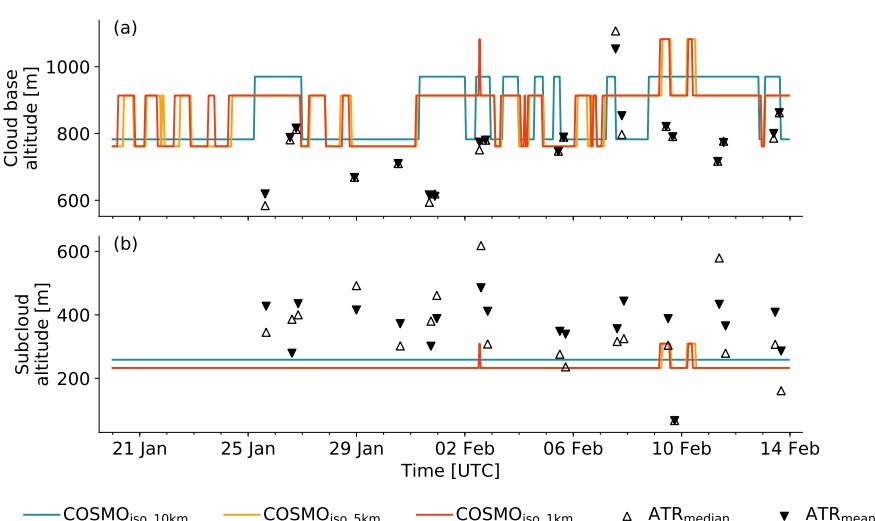

**Figure 3.** Time series of (a) cloud base and (b) subcloud flight altitude. Shown are the median values of the grid points in the domain 54.5-61° W, 11-16° N at every hourly time step for $COSMO_{iso,10km}$ (teal), $COSMO_{iso,5km}$ (yellow), and $COSMO_{iso,1km}$ (red), as well as the median (empty black markers) and mean (filled black markers) value of each ATR flight.



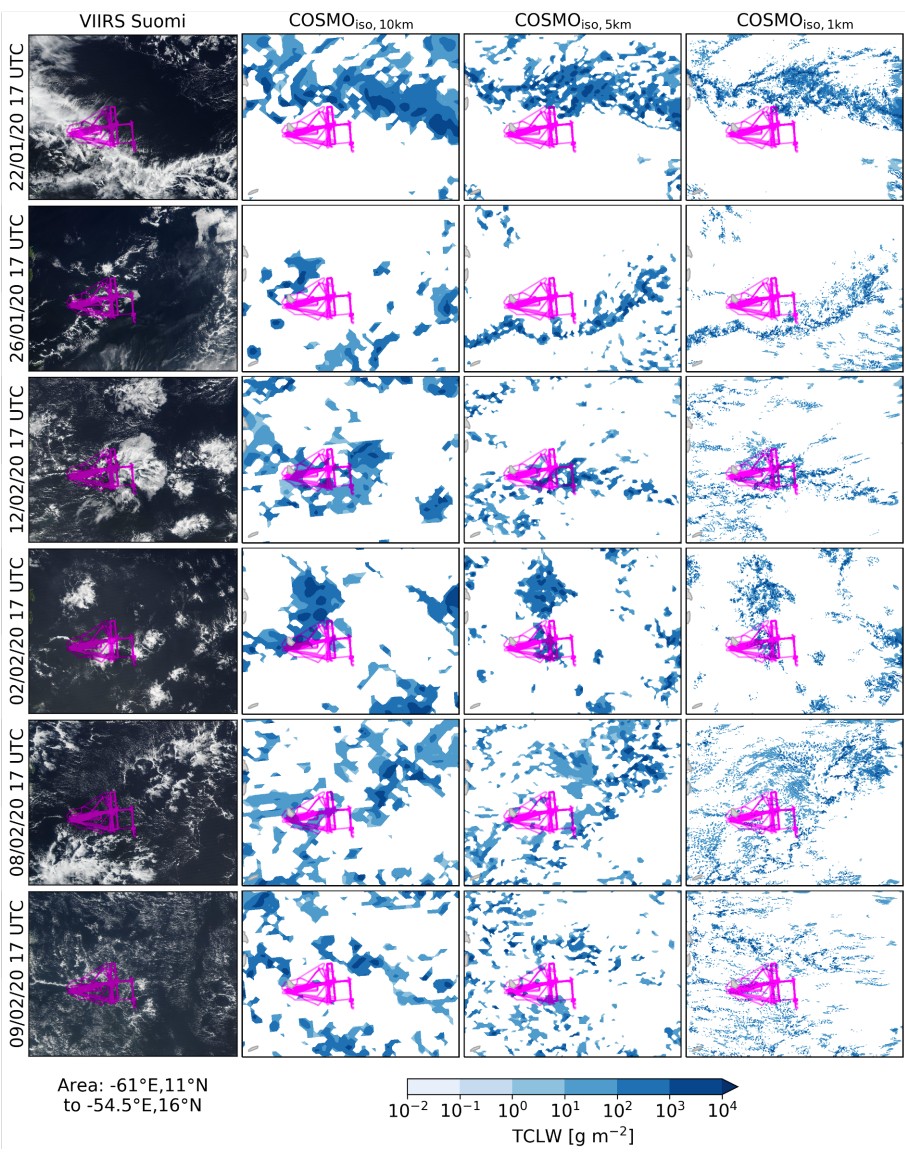

**Figure 4.** Spatial organisation of the clouds in the domain 54.5-61° W, 11-16° N from four datasets (columns) and on six dates (rows). The first column shows the image from the VIIRS instrument onboard the polar-orbiting Suomi-NPP satellite (approx. equatorial crossing time 17:30 UTC). The remaining columns show the total column cloud liquid water (TCLW) in the three COSMO$_{iso}$ simulations at 17:00 UTC. The dates are chosen according to the displayed cloud species, with large/small Fish clouds on 22/26 January 2020, large/small Flower clouds on 12/2 February 2020, and Gravel-Sugar/Sugar clouds on 8/9 February 2020. The track from all ATR flights is overlaid in pink. Other examples are shown in Supplement 1.



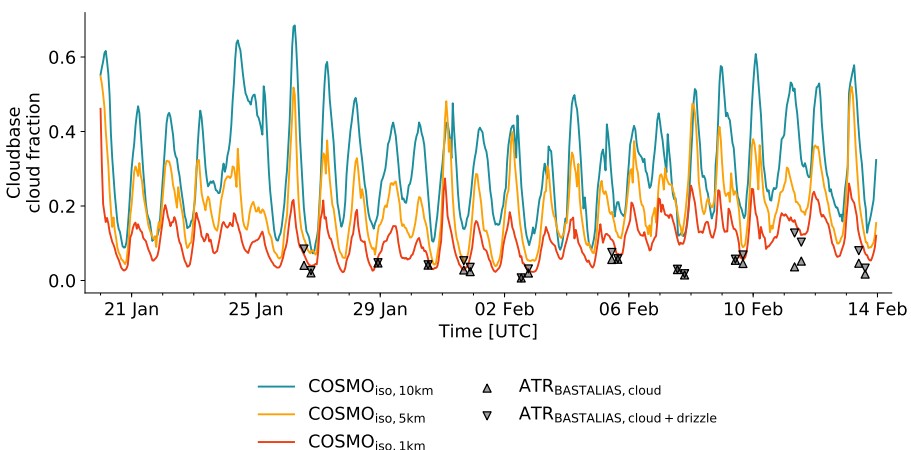

**Figure 5.** Time series of cloud fraction at cloud base for COSMO$_{iso,10km}$ (teal), COSMO$_{iso,5km}$ (yellow), COSMO$_{iso,1km}$ (red) and the ATR (black markers). For COSMO$_{iso}$, the cloud fraction is calculated as the fraction of cloud base grid points in the domain 54.5-61° W, 11-16° N with cloud water content exceeding $10\,\mathrm{mg\,kg^{-1}}$. The number of grid points in the considered domain are 3221 for COSMO$_{iso,10km}$, 12632 for COSMO$_{iso,5km}$, and 316028 for COSMO$_{iso,1km}$.



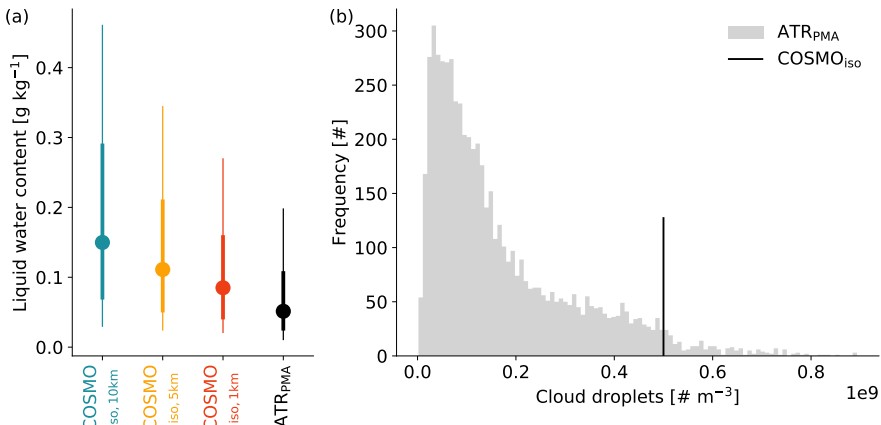

**Figure 6.** Distributions at cloud base of (a) liquid water content (LWC) from COSMO$_{\text{iso,10km}}$ (teal), COSMO$_{\text{iso,5km}}$ (yellow), COSMO$_{\text{iso,1km}}$ (red), and ATR$_{\text{PMA}}$ (black); and (b) cloud droplet numbers from COSMO$_{\text{iso}}$ (black) and ATR$_{\text{PMA}}$ (grey). For COSMO$_{\text{iso}}$, the LWC is calculated as the sum of cloud water content (CWC) and rainwater content (RWC; i.e., LWC = CWC + RWC). Shown are (a) the 10-90-percentile range (thin vertical line), 25-75-percentile range (thick vertical line), and the median (marker) of all cloud base data points with LWC> $10\,\text{mg}\,\text{kg}^{-1}$. For the ATR$_{\text{PMA}}$ dataset, all flights (RF02-RF20) are taken into account, and for the COSMO$_{\text{iso}}$, the data in the domain 54.5-61° W, 11-16° N from the hourly time steps that are closest to the ATR flights. In (b), the droplet number distribution from the ATR$_{\text{PMA}}$ dataset during time steps flagged as cloud is shown together with the constant cloud droplet number of COSMO$_{\text{iso}}$'s one-moment cloud scheme.



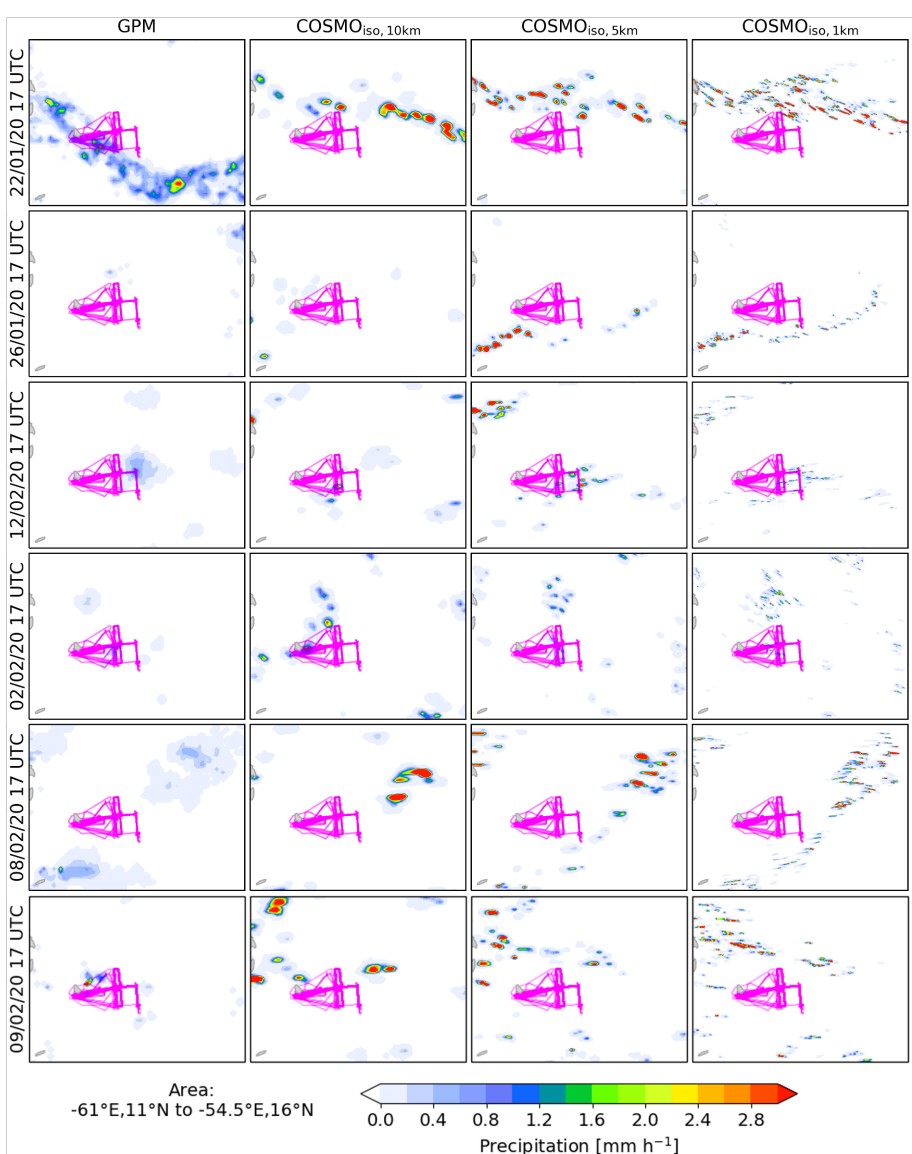

**Figure 7.** Hourly precipitation in the domain 54.5-61° W, 11-16° N from four datasets (columns) and on six dates (rows; also shown in Fig. 4). The first column shows the satellite-based precipitation estimate GPM (IMERG). The remaining columns show precipitation from the three COSMO$_{iso}$ simulations from 16:00-17:00 UTC.



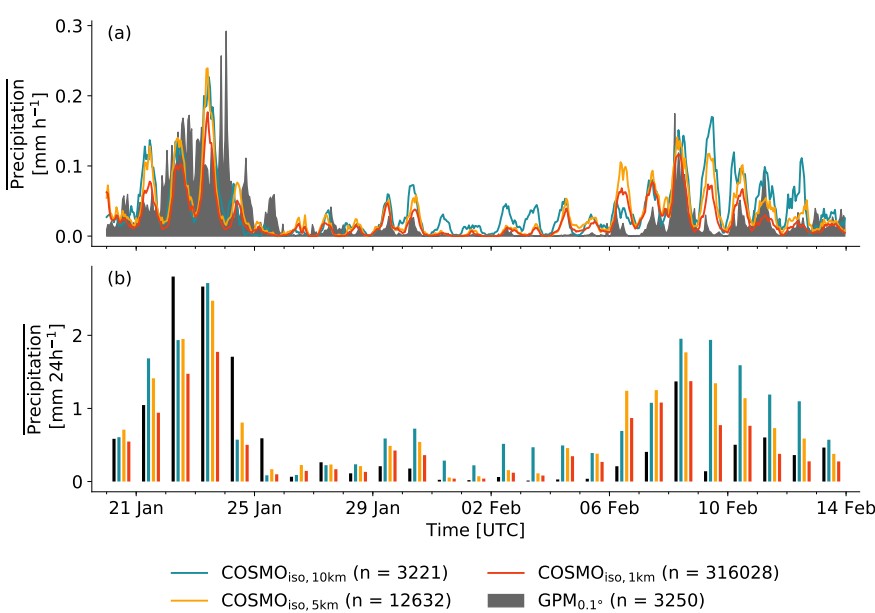

**Figure 8.** Time series displaying (a) hourly and (b) daily domain-averaged precipitation. Shown are the values from COSMO$_{iso,10km}$ (teal), COSMO$_{iso,5km}$ (yellow), COSMO$_{iso,1km}$ (red), and GPM (black). Only the data in the domain 54.5-61° W, 11-16° N are taken into account. The number of grid points in the domain is given in brackets.



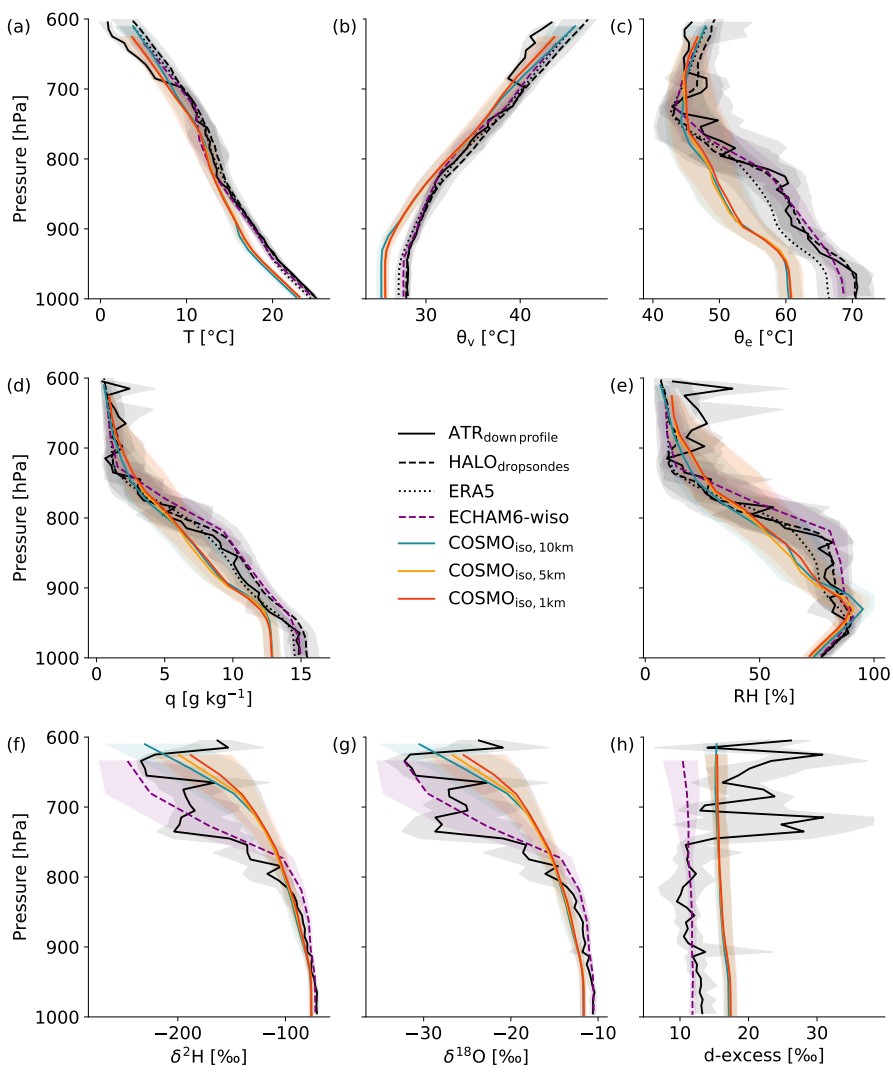

**Figure 9.** Vertical profiles of (a) temperature, (b) virtual potential temperature, (c) equivalent potential temperature, (d) specific humidity, (e) relative humidity, (f) $\delta^2$H in vapour, (g) $\delta^{18}$O in vapour, and (h) d-excess in vapour. Shown are the median (line) and the 25-75-percentile range (shading) of the ATR measurements (black continuous; only the data from the downward profiles from RF03-RF20), the HALO dropsondes (black dashed; 810 sondes), and the vertical profile closest to the centre of the EUREC$^4$A circle (at 57.717° W, 13.3° N) extracted at every hourly time step from 20 January to 13 February 2020 from ERA5 (black dotted; 600 profiles), ECHAM6-wiso (purple dashed; 100 profiles due to the 6-hh time steps), COSMO$_{iso,10km}$ (teal; 600 profiles), COSMO$_{iso,5km}$ (yellow; 600 profiles), COSMO$_{iso,1km}$ (red; 600 profiles). Similar profiles displaying the horizontal wind components are shown in Supplement 2.



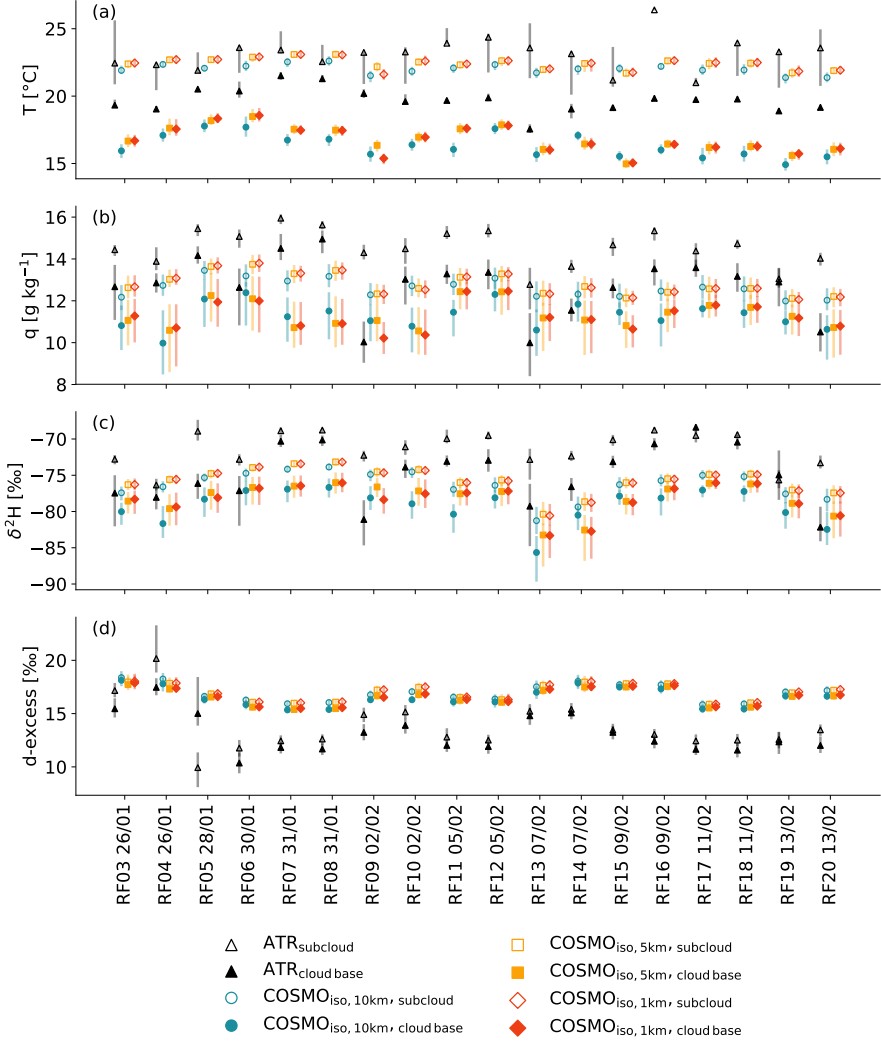

**Figure 10.** Median (marker) and the 25-75-percentile range (transparent line/errorbars) of (a) temperature, (b) specific humidity, (c) $\delta^2$H in vapour, and (d) d-excess for each ATR flight. Shown are the observations from the ATR (black; d-excess, $\delta^2$H, and specific humidity from the Picarro dataset; temperature form the CORE dataset) and the simulations COSMO$_{iso,10km}$ (teal), COSMO$_{iso,5km}$ (yellow), and COSMO$_{iso,1km}$ (red). For the COSMO$_{iso}$ simulations, only the data from the hourly time steps that are closest to the ATR flights, and only the data in the domain 54.5-61° W, 11-16° N are taken into account. Shown are the values for the cloud base (filled markers) and subcloud (empty marker) data points. Note that there is a legend at the bottom of the figure summarizing the dataset and atmospheric levels from which the data points are taken.

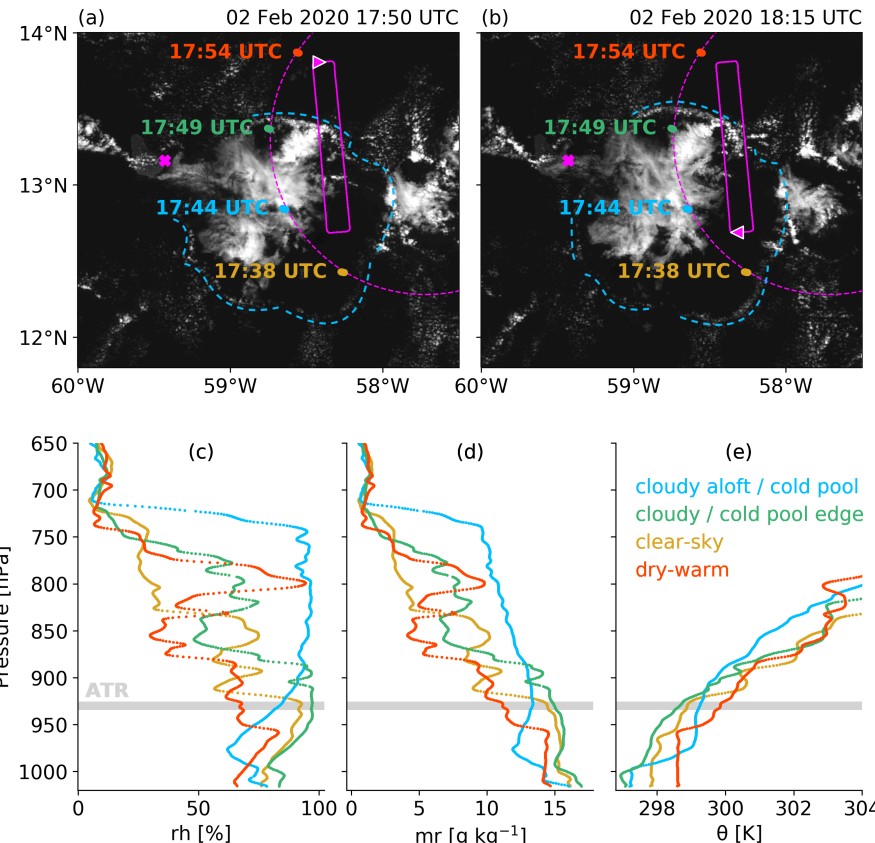

**Figure 11.** Cloudiness viewed by the visible channel of the GOES-16 satellite on 2 February 2020 together with the cloud base flight track of the ATR during RF10 (continuous pink) and the position of the ATR (pink triangle) at (a) 17:50 UTC and (b) 18:15 UTC (time steps chosen such, that the ATR is located at the northern and southern edge of the cloud base rectangle). The two maps also show the horizontal location of four dropsondes launched from the HALO during its circular flight pattern (dashed pink) with the launch time indicated to the left (yellow, blue, green, red). The blue dashed lines show the edges of the cold pool spreading radially away from the cloud in the centre of the satellite image. (c-e) Lower-tropospheric profiles of the four HALO dropsondes (shown in panel a,b) displaying (c) relative humidity, (d) mixing ratio, and (e) potential temperature. Note that the x-axis of (e) is set such that cloud base differences between the profiles are visible. The profile labels in (e) are intended to help the reader link the figure to the text.



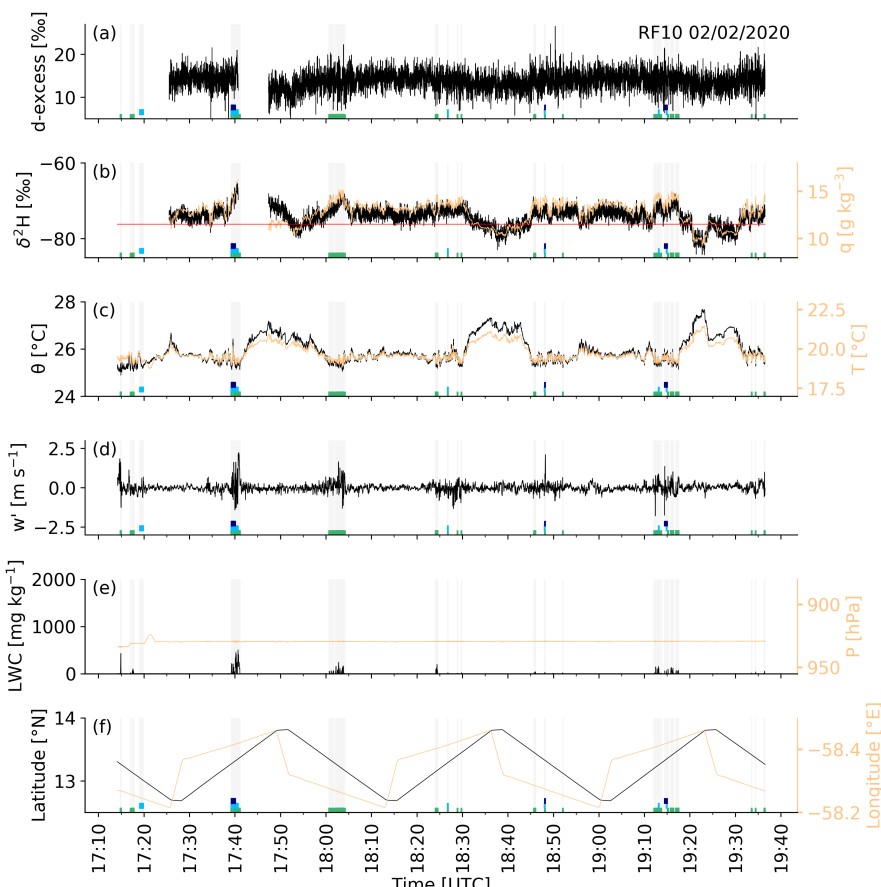

**Figure 12.** Time series of ATR measurements at cloud base during RF10 on 2 February 2020. Shown are (a) d-excess in vapour, (b) $\delta^2$H in vapour (black) and specific humidity (sandy), (c) potential temperature (black) and temperature (sandy), (d) vertical wind velocity fluctuations relative to the flight-segment mean, (e) liquid water content (black) and pressure (sandy), (f) latitude (black) and longitude (sandy) of the ATR's position. In all panels, the grey shading indicates the presence of liquid water and the green/lightblue/darkblue bars at the bottom of (a,b,c,d,f) the cloud/drizzle/rain flags from the PMA dataset. Temperatures, vertical velocity, and positional data come from the ATR CORE dataset; d-excess, $\delta^2$H, and specific humidity from the ATR Picarro dataset; and liquid water content from the ATR PMA dataset.



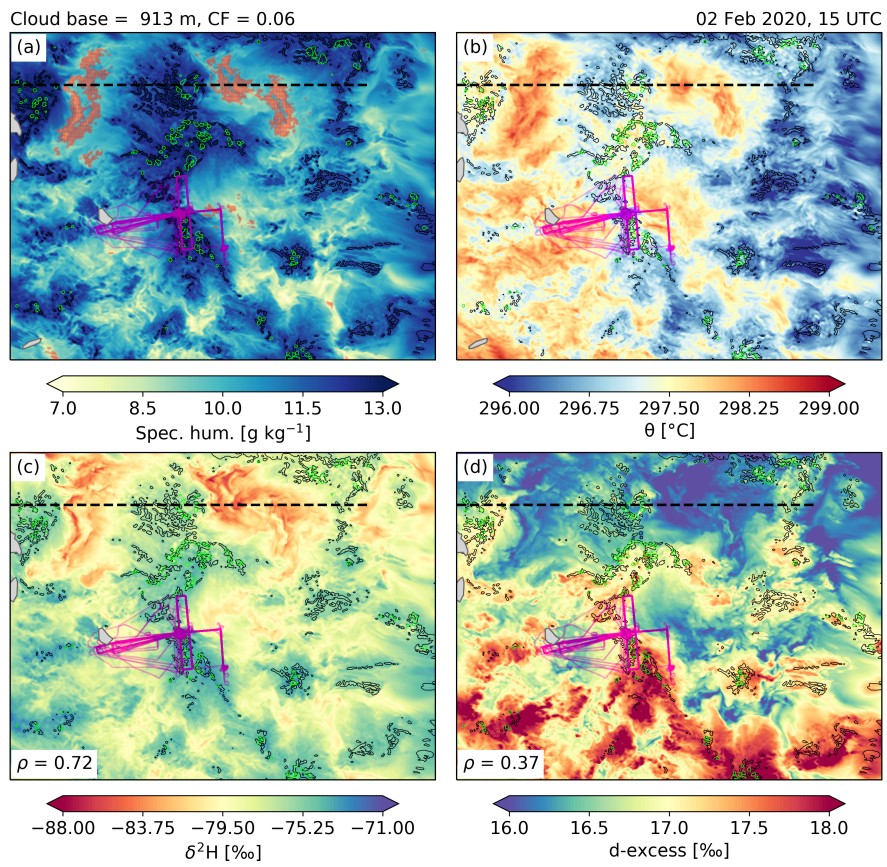

**Figure 13.** Fields of (a) specific humidity, (b) potential temperature, (c) $\delta^2$H in vapour, and (d) d-excess in vapour at cloud base (z=913 m) in the domain 54.5-61° W, 11-16° N from the COSMO$_{iso,1km}$ dataset at 15:00 UTC on 2 February 2020. The transparent red dots in (a) show grid points identified as *dry-warm* (see definition in text); the pink lines represent the track from all ATR flights; and the black line indicates the position of the crosssection shown in Fig. 14. Liquid cloud water contents of $10\,\mathrm{mg\,kg^{-1}}$ are shown as black contours and rain water contents of $1\,\mathrm{mg\,kg^{-1}}$ as green contours. In (c,d), the Pearson correlation coefficients ($\rho$) between the cloud base specific humidity and (c) $\delta^2$H in vapour, (d) d-excess in vapour are indicated in the lower left corner. The cloud base cloud fraction (CF) at this time and in this region corresponds to 6 %.



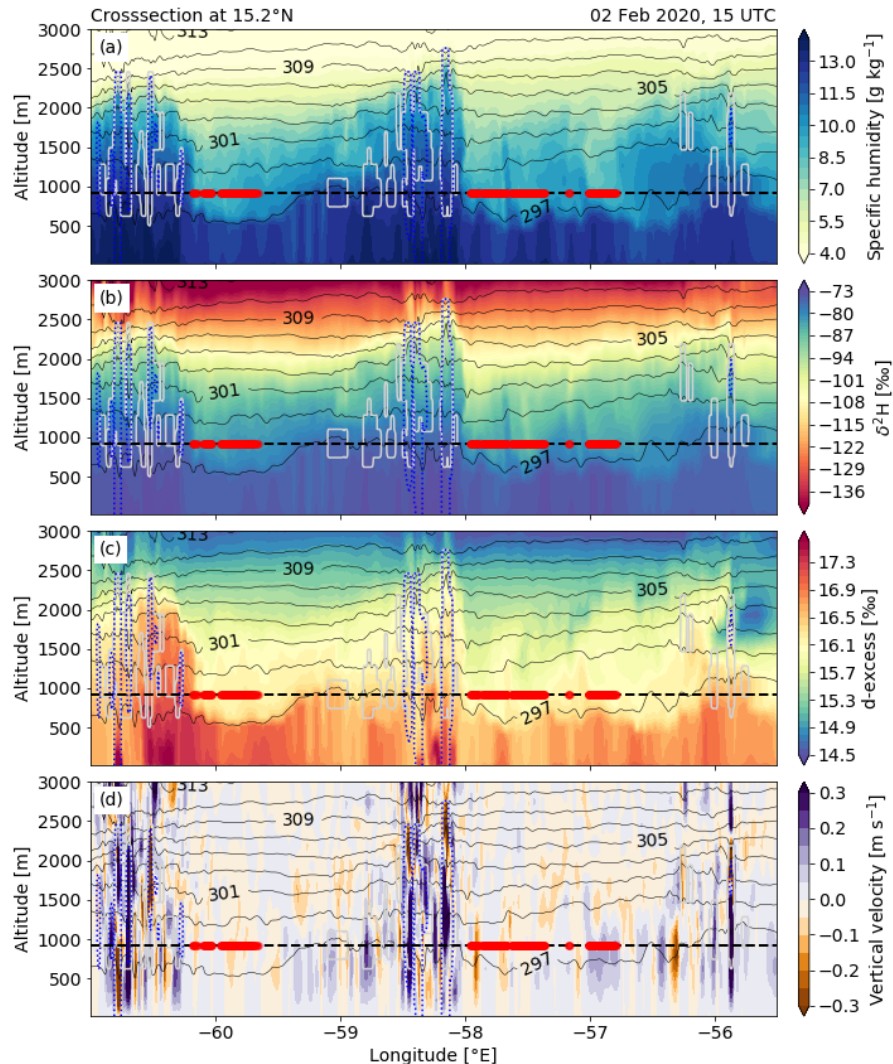

**Figure 14.** Crosssections at $15.2°$ N of (a) specific humidity, (b) $\delta^2$H in vapour, (c) d-excess in vapour, and (d) vertical velocity from COSMO$_{iso,1km}$ at 15:00 UTC on 2 February 2020. Shown is also potential temperature (black contours), cloud water contents of $10\,\mathrm{mg\,kg^{-1}}$ (grey contours), rain water contents of $10\,\mathrm{mg\,kg^{-1}}$ (dashed blue contours) and cloud base height (dashed black line; the level at which the fields are shown in Fig. 13) with the cloud base grid points identified as *dry-warm* (red dots; less than $0.01°$ latitude away from the crosssection). The location of the crosssection is indicated in Fig. 13.





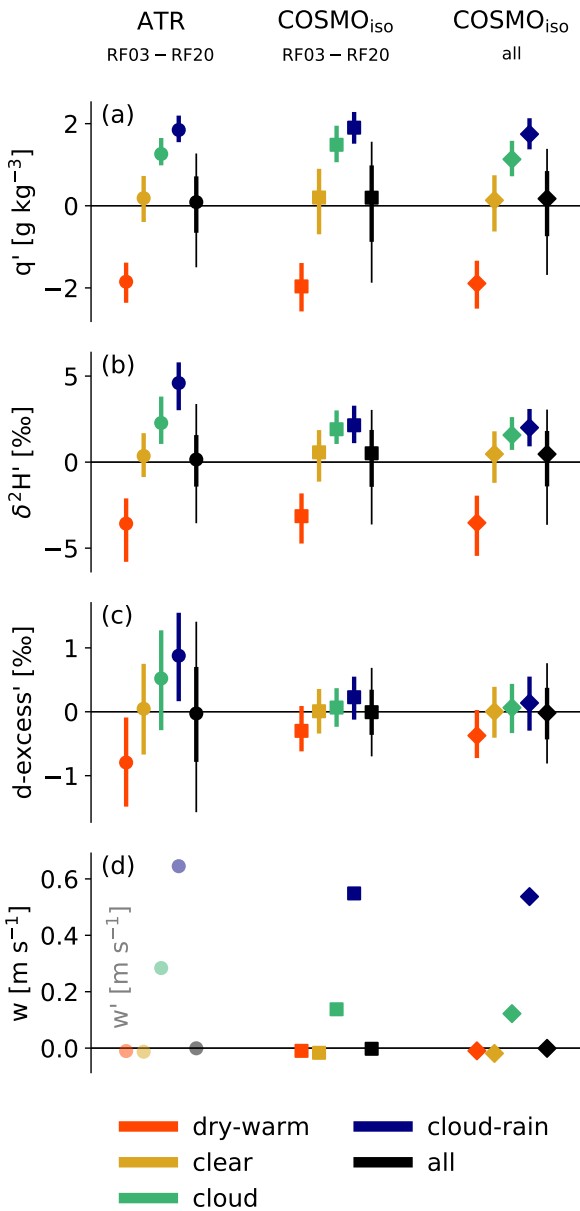

**Figure 15.** Distributions of the cloud base anomalies in (a) specific humidity, (b) $\delta^2$H in vapour, and (c) d-excess in vapour of the data points identified as *dry-warm*, *clear*, *cloud*, *cloud-rain* (see the text for the definition) and all of them together (black). For the ATR data, anomalies are defined relative to each flight's mean cloud base conditions. For COSMO$_{iso,1km}$, anomalies are defined relative to each time steps mean cloud base conditions in the domain 54.5-61° W, 11-16° N. Shown are the median (markers), the 25-75-percentile range (lines), and for the overall distribution (black) the 10-90-percentile range (thin black line) of the cloud base data points collected by the ATR (RF03-RF20) and the COSMO$_{iso,1km}$ cloud base grid points at the hourly time steps that are closest to the ATR flights (COSMO$_{iso,RF03-RF20}$) and over the whole simulated period (20 January to 13 February 2020; COSMO$_{iso,all}$). (d) The mean absolute vertical velocity $w$ for COSMO$_{iso}$ and mean vertical velocity fluctuations $w'$ for the ATR (see text for the reasoning behind the variable choice) at cloud base. For the ATR, only RF10-RF19 provide vertical velocity measurements and are included here. Table 5 gives the percentage of data points in each category and Table 6 numerical values of the median anomalies. Supplement 3 provides alternative versions of this figure.



**Table 1.** Model setup for the ECHAM6-wiso, COSMO$_{iso,10km}$, COSMO$_{iso,5km}$, and COSMO$_{iso,1km}$ simulations. The following abbreviations are used to describe the variables included in the nudging: surface pressure ($SP$), temperature ($T$), zonal wind component ($U$), and meridional wind component ($V$).

|  | ECHAM6-wiso | COSMO$_{iso,10km}$ | COSMO$_{iso,5km}$ | COSMO$_{iso,1km}$ |
|---|---|---|---|---|
| Resolution | T127 (ca. 0.9°) | 0.1° (ca. 10km) | 0.05° (ca. 5km) | 0.01° (ca. 1km) |
| Grid centre | - | 45° W, 25° N | 57.7° W, 13.3° N | 57.7° W, 13.3° N |
| Grid points | $193 \times 139 \times 95$ | $900 \times 800 \times 40$ | $700 \times 600 \times 60$ | $700 \times 600 \times 60$ |
| Start | 1 Dec 2019 | 6 Jan 2020 | 20 Jan 2020 | 20 Jan 2020 |
| End | 29 Feb 2020 | 13 Feb 2020 | 13 Feb 2020 | 13 Feb 2020 |
| Time steps | 6 h | 1 h | 1 h | 1 h |
| Convection | Parameterised | Explicit | Explicit | Explicit |
| Boundary | ERA5 | ECHAM6-wiso | COSMO$_{iso,10km}$ | COSMO$_{iso,5km}$ |
| Nudging | $SP, T, U, V$ | $U,V$ | $U,V$ | $U,V$ |





**Table 2.** Root mean square differences of the median profiles (shown in Fig. 9) of the HALO dropsondes, ERA5, ECHAM6-wiso, COSMO$_{iso,10km}$, COSMO$_{iso,5km}$, and COSMO$_{iso,1km}$ data relative to the ones from the ATR$_{down\ profile}$ data over the layer 1000-850 hPa.

| Dataset | $T$ [°C] | $\theta_v$ [°C] | $\theta_e$ [°C] | $q$ [g kg$^{-1}$] | $RH$ [%] | $\delta^2$H [‰] | $\delta^{18}$O [‰] | $d$ [‰] |
|---|---|---|---|---|---|---|---|---|
| HALO | 0.1 | 0.1 | 0.8 | 0.8 | 2.8 | - | - | - |
| ERA5 | 0.5 | 0.6 | 3.3 | 0.4 | 3.9 | - | - | - |
| ECHAM6-wiso | 0.4 | 0.3 | 1.6 | 0.8 | 4.7 | 3.1 | 0.4 | 1.0 |
| COSMO$_{iso,10km}$ | 2.1 | 2.5 | 8.7 | 1.6 | 8.4 | 4.6 | 1.1 | 4.4 |
| COSMO$_{iso,5km}$ | 1.9 | 2.2 | 8.6 | 1.7 | 8.7 | 3.9 | 1.0 | 4.6 |
| COSMO$_{iso,1km}$ | 1.8 | 2.2 | 8.5 | 1.6 | 6.8 | 3.7 | 1.0 | 4.6 |



**Table 3.** Pearson correlation coefficients for temperature ($T$), specific humidity ($q$), $\delta^2 H$ in vapour, and d-excess in vapour ($d$) between the median values from each ATR flight (RF03-RF20) and the hourly median values of all grid points in the domain 54.5-61° W, 11-16° N that are closest to the ATR flights from the three COSMO$_{\text{iso}}$ simulations. Shown are cloud base and subcloud (in brackets) values.

| Dataset | $T$ [°C] | $q$ [g kg$^{-1}$] | $\delta^2$H [‰] | $d$ [‰] |
|---|---|---|---|---|
| COSMO$_{\text{iso,10km}}$ | 0.44 (0.03) | 0.35 (0.73) | 0.65 (0.50) | 0.74 (0.73) |
| COSMO$_{\text{iso,5km}}$ | 0.56 (0.13) | 0.25 (0.65) | 0.60 (0.46) | 0.72 (0.66) |
| COSMO$_{\text{iso,1km}}$ | 0.50 (0.11) | 0.34 (0.66) | 0.69 (0.45) | 0.72 (0.66) |



**Table 4.** Root mean square differences between the subcloud and the cloud base values for temperature ($T$), specific humidity ($q$), $\delta^2$H in vapour, and d-excess in ($d$) vapour between the median values from each ATR flight (RF03-RF20) and the hourly median value of all grid points in the domain 54.5-61° W, 11-16° N that are closest to the ATR flights from the three COSMO$_{iso}$ simulations.

| Dataset | $T$ [°C] | $q$ [g kg$^{-1}$] | $\delta^2$H [‰] | $d$ [‰] |
|---|---|---|---|---|
| COSMO$_{iso,10km}$ | 5.7 | 1.4 | 3.1 | 0.5 |
| COSMO$_{iso,5km}$ | 5.7 | 1.6 | 2.6 | 0.5 |
| COSMO$_{iso,1km}$ | 5.7 | 1.6 | 2.7 | 0.4 |
| ATR | 3.8 | 2.1 | 4.5 | 1.7 |



**Table 5.** Percentage of data points assigned to one of the four cloud base categories (*dry-warm*, *clear*, *cloud*, *cloud-rain*). For the ATR, the time steps with valid isotope data are taken into account (RF03-RF20; note that these are fewer data points than available in the PMA dataset alone, because the isotope dataset has data gaps due to calibration or instrument failure, especially during rainy flight segments). For the COSMO$_{iso,1km}$, the data in the domain 54.5-61° W, 11-16° N is taken into account; once only from the hourly time steps that are closest to the ATR flights (COSMO$_{iso,RF03-RF20}$), and once for all simulated time steps (COMSO$_{iso,all}$).

| Dataset | *dry-warm* | *clear* | *cloud* | *cloud-rain* |
|---|---|---|---|---|
| ATR | 11.7 % | 84.3 % | 3.8 % | 0.2 % |
| COSMO$_{iso,RF03-RF20}$ | 8 % | 84 % | 7.7 % | 0.3 % |
| COMSO$_{iso,all}$ | 7.6 % | 81.2 % | 10.9 % | 0.3 % |



**Table 6.** Median values of the cloud base anomalies in specific humidity ($q$), $\delta^2$H in vapour, and d-excess ($d$) in vapour for the data points identified as *dry-warm*, *clear*, *cloud*, *cloud-rain* (see the text for the definition) and the difference between the median of the cloud-rain and the dry-warm grid points (calculated with the exact values and not the rounded medians shown here) from the ATR and COSMO$_{iso,1km}$. For the ATR data, anomalies are defined relative to each flight's mean cloud base conditions. For COSMO$_{iso,1km}$, anomalies are defined relative to each time step's hour-of-the-day average cloud base conditions in the domain 54.5-61° W, 11-16° N. Shown are the median values over the cloud base time steps of the ATR (RF03-RF20), the COSMO$_{iso,1km}$ cloud base grid points at the hourly time steps that are closest to the ATR flights (COSMO$_{iso,RF03-RF20}$) and over the whole simulated period (20 January to 13 February 2020; COSMO$_{iso,RF03-RF20}$). Note that the 25-50-75 percentiles of these variables in each cloud base environment are shown in Fig. 15.

| Dataset | Variable | *dry-warm* | *clear* | *cloud* | *cloud-rain* | *cloud-rain* minus *dry-warm* |
|---|---|---|---|---|---|---|
| ATR | | −1.8 | 0.2 | 1.3 | 1.8 | 3.6 |
| COSMO$_{iso,RF03-RF20}$ | $q$ [g kg$^{-1}$] | −2.0 | 0.2 | 1.5 | 1.9 | 3.9 |
| COMSO$_{iso,all}$ | | −1.9 | 0.1 | 1.1 | 1.7 | 3.6 |
| ATR | | −3.6 | 0.4 | 2.3 | 4.6 | 8.2 |
| COSMO$_{iso,RF03-RF20}$ | $\delta^2$H [‰] | −3.1 | 0.6 | 1.9 | 2.1 | 5.2 |
| COMSO$_{iso,all}$ | | −3.5 | 0.5 | 1.6 | 2.0 | 5.5 |
| ATR | | −0.8 | 0.0 | 0.5 | 0.9 | 1.7 |
| COSMO$_{iso,RF03-RF20}$ | $d$ [‰] | −0.3 | 0.0 | 0.1 | 0.2 | 0.5 |
| COMSO$_{iso,all}$ | | −0.4 | 0.0 | 0.1 | 0.1 | 0.5 |