# Peer review of "Water isotopic characterisation of the cloud-circulation coupling in the North Atlantic trades. Part 1: A process-oriented evaluation of COSMOiso simulations with EUREC4A observations"

_EGUsphere, 2023_

## Referee Comment (RC1)

**Review of Villiger et al, part 1**

April 17, 2023

This article presents an evaluation of high-resolution isotope-enabled simulations of the North Atlantic trades against observations from the EUREC4A campaign. This evaluation serves a a basis for the part 2 of the study, which investigates the role of the atmospheric circulation at different spatial scales on the isotopic variability. This article also presents interesting elements in itself, such as the sensitivity of the results to horizontal resolution and the isotopic signatures of clouds and of dry-warm patches.

The article is well written, although it is quite long. It is well illustrated. I have only minor comments.

- l 16: "mesoscale cloud base feature" is a bit enigmatic at this stage. Maybe be more specific, e.g. isotopic characteristics were investigated under precipitating clouds, non-precipitating clouds, clear-sky conditions and dry-warm patches.

- l 18: I don't understand what these numbers represent: what do the bounds represent? And the numbers in brackets? Maybe clarify with one or two additional sentences.

- l 313: clarify in this sentence if this is a daily value, or a temporal-mean? Or maybe it doesn't matter because the closest level to the mid-level is constant in time?

- p10, Fig 3:
    - How were cloud-base and subcloud levels defined in observations? A few sentences on this would be helpful.
    - Why is the subcloud level defined as mid-level between cloud base and the surface in simulations? COSMO overestimates the cloud-base level but systematically underestimates the subcloud level. Why not defining the subcloud level in COSMO in a way that is more consistent with observations?
    - To what extent do the variations in cloud-based altitudes contribute to the variations of observed and simulated variables at this altitude?
    - To what extent do the model-data differences in cloud-based altitudes contribute to the model-data differences in variables at this altitude?

- l 325: unclear sentence. Do you mean "... is assessed, without upscaling outputs from the higher resolution simulations to the lowest resolution"?

- l 418: "derive from it" -> "conclude"?

- l 420: "A behavior also" -> "This behavior is also"

- l 462: "10‰" -> "10‰ in $\delta^2 H$"?

- l 464: "promoting too" -> "promoted by too"? I understand from section 3.1 that the number of cloud droplets and their size is determined by some tunable parameter, so the overestimated evaporation is a consequence of this, rather than the reverse?

- l 460-467: What is the difference between the two mechanisms? It sounds like they are equivalent. In both cases, overestimated cloud evaporation leads to weak moistening and strong enrichment. The underestimated precipitation efficiency in (1) sounds like just a bulk view to simply formalize this effect, whereas (2) is the detailed mechanism?

- l 477-479: the impact of the low-level dry bias on the high bias in d-excess could easy be quantified: the slope of d-excess as a function of surface relative humidity is easy to estimate under the Merlivat and Jouzel 1979 approximation. If the low-level dry bias is enough to explain the high bias in d-excess, then it is not necessary to invoke more complicated explanations.

- l 636: the role of "partial evaporation of falling hydrometeors" is discussed, and then the conclusion is on "mesoscale transport processes". Doesn't the variability in d-excess rather reflect a combination of microphysical and mesoscale transport processes?

- l 668: "the higher the" -> "at higher"?

- l 706: "linked to shallow convective upward transport": and partial evaporation of falling hydrometeors?

- Fig 5:
  - To what extent is the simulated cloud-base cloud fraction sensitive to the threshold on condensed water content? Is there a possibility that the cloud fraction in observations and in the simulations represent two different things?
  - Is it possible that the sensitivity of the cloud fraction to the resolution is just an artifact of the different resolutions? Cloud fraction can be very sensitive to the resolution at which a cloudy pixel is defined. If all simulation outputs were coarse-grained on the same grid, would they exhibit such a large difference in cloud fraction? In other words, is the large difference related to different physical states, or to different definitions of a cloud? What does the observed cloud fraction represent, in terms of horizontal resolution of what is defined a cloud?

---

## Author Comment (AC1)

**Authors' response to reviews of the manuscript**

**"Water isotopic characterisation of the cloud-circulation coupling in the North Atlantic trades. Part 1: A process-oriented evaluation of COSMO$_{\mathrm{iso}}$ simulations with EUREC$^4$A observations"**

by Leonie Villiger et al.

We thank the anonymous reviewers for their insightful comments, which we address in detail in our responses below. They helped us to further improve the presentation of our results and led to the following main changes in the revised manuscript:

1. We added several statements to better justify the use of water isotopes for a non-isotope readership.

2. We shortened the paper:

   - by removing the analysis of the subcloud layer variables,
   - by removing the temperature and specific humidity from the comparison between cloud base ATR observations and COSMO$_{\mathrm{iso}}$ data (Fig. 10 of the paper) because these two variables were shown to be strongly influenced by the cloud base altitude,
   - by shortening the explanation on the d-excess bias.

3. We reformulated the abstract and Sect. 5 (Summary and conclusion) to clarify that (1) we expect a resolution-dependency in cloud-related variables due to the subgrid-scale nature of shallow cumulus clouds in the range of 1-10 km and (2) the primary goal of the study is to assess whether the presented simulations can be used for studying cloud processes in the lower troposphere of the trade-wind region, rather than advancing the realm of model development.

Please find our detailed response to the reviewers' comments below in the following format:
Reviewer's comment. Authors' response.
* * *
**Reviewer 1:**

This article presents an evaluation of high-resolution isotope-enabled simulations of the North Atlantic trades against observations from the EUREC4A campaign. This evaluation serves as a basis for the part 2 of the study, which investigates the role of the atmospheric circulation at different spatial scales on the isotopic variability. This article also presents interesting elements in itself, such as the sensitivity of the results to horizontal resolution and the isotopic signatures of clouds and of dry-warm patches.
We thank reviewer 1 very much for her/his insightful comments, which were very valuable for improving the presentation of this study.

**Minor comments:**

1.1 The article is well written, although it is quite long. It is well illustrated. I have only minor comments.
We thank reviewer 1 for this overall positive feedback. Based on the remark that the article is quite long, a concern shared with reviewer 2 (see comment A), we decided to shorten the manuscript by removing the analyses discussing the subcloud layer data (observations and simulations). We have thus removed:

   - the description of the subcloud-layer level definition from Sect. 2.3 and changed the title of this subsection from "*Definition of relevant atmospheric levels*" to "*Definition of the cloud base level*",
   - the subcloud-layer data points from Fig. 10 and all related text,
   - the correlation coefficients for the subcloud-layer data points in Table 3 and all related text,

- Table 4 and all related text.

These findings are not crucial for the key messages of the study and we refer readers to Villiger (2022), in which similar analyses can be found. In addition, we have shortened text where possible without loosing relevant information (e.g., bracket comments that repeat content from the main text).

1.2 L16: "mesoscale cloud base feature" is a bit enigmatic at this stage. Maybe be more specific, e.g. isotopic characteristics were investigated under precipitating clouds, non-precipitating clouds, clear-sky conditions and dry-warm patches.

This has been adapted as suggested. The new sentence is:

*"A close examination of isotopic characteristics under precipitating clouds, non-precipitating clouds, clear-sky and dry-warm patches at the altitude of cloud base shows that these different environments are represented faithfully in the model with similar frequency of occurrence, isotope signals and specific humidity anomalies as found in the observations."*

1.3 L18: I don't understand what these numbers represent: what do the bounds represent? And the numbers in brackets? Maybe clarify with one or two additional sentences.

The numbers (based on Table 6) represent the cloud base anomalies of $\delta^2$H and specific humidity (given in square brackets) in dry-warm patches at the cloud base level and below precipitating clouds. For specific humidity, the (rounded) anomalies in the observations are identical to the ones in the simulation, which is why only one number was given for each of the two cloud base environments. For $\delta^2$H the anomalies differ between the observations and the simulations, which is why two numbers for each of the cloud base environments were given. For clarity, we have simplified the sentence and no longer state the explicit values of the anomalies. The text has been adapted as follows:

*"A close examination of isotopic characteristics under precipitating clouds, non-precipitating clouds, clear-sky and dry-warm patches at the altitude of cloud base shows that these different environments are represented faithfully in the model with similar frequency of occurrence, isotope signals and specific humidity anomalies as found in the observations. Furthermore, it is shown that the $\delta^2 H$ of cloud base vapour at the hourly timescale is mainly controlled by mesoscale transport and not by local microphysical processes, while the d-excess is mainly controlled by large-scale drivers."*

1.4 L313: clarify in this sentence if this is a daily value, or a temporal-mean? Or maybe it doesn't matter because the closest level to the mid-level is constant in time?

This paragraph about the subcloud layer has been removed (see comment 1.1). In the original text, we meant a temporal mean. The subcloud layer level varied slightly in time because it was defined as the mid-level between the surface and cloud base level, which was defined separately for each hourly time step.

1.5 p10, Fig 3:

- How were cloud-base and subcloud levels defined in observations? A few sentences on this would be helpful.
    - **Cloud-base level:** Bony et al. (2022, p. 2025) describes in detail how cloud base was determined: *"[...] the targeted cloud base level was not the lifting condensation level (LCL) but the height of the maximum near-base cloud fraction ($z_{CFmax}$). This level corresponds to the level where most clouds in the sampling area have reached their base level, and it is most adequately defined by the height at which a cloud radar reports a maximum cloudiness near cloud base. The cloud base height distributions from the ceilometer and estimates of the mixed-layer-top, subcloud-layer-top (h) and LCL from soundings and surface weather data provided further guidance for choosing the correct cloud base level. The evening before the flight, and again 2 h before take-off, a pre-flight estimation of the flight levels was performed based on near-real-time cloud radar, ceilometer, radiosonde and surface weather data from the BCO and R/V Meteor as well as satellite imagery [...]. During the flights, real-time ATR lidar backscatter quick-looks and visual impressions from the pilots, as well as real-time information from the HALO dropsondes and lidar quick-looks [...], were used to fine-tune the flight level."*

    We have added the following summary of this information in Sect. 2.3:

    *"The targeted cloud base level was determined before each flight based on ceilometer, radio*

*sounding, and surface weather observations as well as satellite imagery. During the flight, the altitude was fine-tuned based on real-time measurements and visual impressions."*

– **Subcloud level:** All flight segments below cloud base and above the surface legs were labelled as subcloud layer. Consequently, subcloud levels flown by the ATR range from $\sim 150$ to $\sim 600\,\mathrm{m}$ (Bony et al., 2022). However, since the discussion about the subcloud layer observations was removed, we did not add this additional information to the text.

- Why is the subcloud level defined as mid-level between cloud base and the surface in simulations? COSMO overestimates the cloud-base level, but systematically underestimates the subcloud level.
  Panel b) of Fig. 3 has been removed, therefore, this comment becomes irrelevant for the revised version of the paper. Nevertheless, for the reviewer's personal interest, the subcloud level was defined as mid-level between cloud base and the surface building on the fact that the subcloud layer is well-mixed (see vertical profiles shown in Fig. 9 of the paper), except for the thin transition layer at its top (Albright et al., 2022). In other words, the exact altitude (i.e., the subcloud level) at which the subcloud layer is sampled doesn't matter as long as it is well away (i.e., at least one model level) from the cloud base level. By choosing the mid-level between cloud base and the surface, we ensure to sample the subcloud layer.
  Alternative approaches of defining the subcloud level could be to extract the model level, which is closest to the subcloud altitudes of the ATR (and interpolate between time steps with no ATR flights) or extracting the model level closest to the mean subcloud altitude of the ATR. However, we do not expect different values at these levels than at the mid-level (due to the well-mixed characteristics of the subcloud layer). Moving away from the idea of extracting subcloud layer variables at one specific level, one could also have compared subcloud layer mean values of the simulations (i.e., mean properties between the surface and cloud base level) with the subcloud layer observations of the ATR. However, we do not see an apparent advantage of this approach over the one applied in the thesis (Villiger, 2022).

- Why not defining the subcloud level in COSMO in a way that is more consistent with observations?
  See response to previous comment.

- To what extent do the variations in cloud-based altitudes contribute to the variations of observed and simulated variables at this altitude?
  The variations in cloud-base altitudes do contribute to the variations of observed and simulated variables (Fig. R1). In the simulations, we find a linear dependency of cloud base temperature (Fig. R1a) and specific humidity on cloud base altitude (Fig. R1b). For cloud base $\delta^2\mathrm{H}$ (Fig. R1c) or d-excess (Fig. R1d), however, no dependency on the cloud base altitude is found. In the ATR data (with more continuous cloud base altitudes compared to the simulations), we find linear dependencies for all four cloud base variables on cloud base altitude, with the strongest for temperature (Fig. R1a), followed by specific humidity (Fig. R1b), $\delta^2\mathrm{H}$ (Fig. R1c), and only a weak one for d-excess (Fig. R1d). The fact that the isotope variables are less sensitive to the altitude can also be seen in the vertical profiles shown in Fig. 9f-h of the paper. It is important to note that for our analysis of the contrasting isotope signals in different cloud base environments (i.e., precipitating clouds, non-precipitating clouds, dry-warm patches, and clear-sky patches; Fig. 15 of the paper), these dependencies do not matter because we only consider anomalies relative to cloud base level mean conditions of each time step (in case of the simulations) or of each flight (in case of the ATR data).

[Figure]

**Figure R1:** Relation between cloud base altitude and cloud base **(a)** temperature, **(b)** specific humidity, **(c)** $\delta^2$H, and **(d)** d-excess for COSMO$_{iso,1km}$ (red diamonds) and the ATR (black triangles) data. For COSMO$_{iso,1km}$, the median cloud base value in the domain 54.5-61° W, 11-16° N for each hourly time step is shown. For the ATR, the median value for each flight (RF03-RF20) is shown. The number in the upper right corner of each panel represents the Pearson correlation coefficient between the respective $x$ and $y$ variable, and the dotted line the linear regression line (red for the COSMO$_{iso,1km}$ data and black for the ATR data).

- To what extent do the model-data differences in cloud-base altitudes contribute to the model-data differences in variables at this altitude?

  We are not sure what is meant with *model-data differences*, but assume that it refers to the differences between the simulations and the ATR observations, which therefore also relates to the discussion of Fig. 10 in the paper. The differences between simulation and ATR cloud base data show a strong negative correlation between altitude and temperature (Fig. R2a, corr. = -0.75) as well as altitude and specific humidity (Fig. R2b, corr. = -0.65). In other words, the differences in these two cloud base variables between COSMO$_{iso}$ and the ATR observations partly reflect the differences in cloud base altitudes. Nevertheless, a clear cold bias can be found in the simulations (i.e., $\Delta T$ in Fig. R2a are all negative regardless of whether $\Delta$Cloud base altitude is negative or positive); its magnitude simply scales with the differences in cloud base altitude. For specific humidity, the picture is somewhat less clear (Fig. R2b; most but not all $\Delta$q are negative regardless of whether $\Delta$Cloud base altitude is positive or negative). For $\delta^2$H (Fig. R2c) and d-excess (Fig. R2c), there is no clear scaling of the differences in the variable with the differences in cloud base altitudes.

  From this analysis, we conclude that addressing the low-level biases of variables which show a strong altitude dependence (i.e., temperature and specific humidity) at cloud base by evaluating their flight-to-flight variability at cloud base might be confusing, since they largely reflect the flight-to-flight variability of the cloud base altitude. Based on this analysis, we realised that the low correlations between simulations and observations for temperature and specific humidity are due to the strong altitude dependency of these variables (which is larger in the

ATR data than the COSMO$_{iso}$ data due to the larger range of flown altitudes). As a consequence, we have removed temperature and specific humidity from Fig. 10 and Table 3 of the paper and have adapted the paragraph discussing the two variables to the following:

*"After having discussed various biases, we want to evaluate the development of isotope variables over time. To this end, we focus on cloud base and compare the flight-to-flight variability found in the ATR data to the simulations. High correlations (Table 3) between the two datasets (Fig. 10) suggest that the observed temporal evolution of $\delta^2 H$ and d-excess is well captured by the three COSMO$_{iso}$ simulations. Another similarity between observations and simulations is that $\delta^2 H$ (Fig. 10 a) generally shows a larger in-flight spatial variability at cloud base compared to the flight-to-flight temporal (synoptic) variability. This highlights the importance of mesoscale variability at cloud base, which is the topic of the following Sect. 4."*

[Figure]

**Figure R2:** Relation between differences (COSMO$_{iso,1km}$ - ATR) of cloud base altitude and cloud base values of **(a)** temperature, **(b)** specific humidity, **(d)** $\delta^2$H, and **(d)** d-excess. For COSMO$_{iso,1km}$, the median cloud base value in the domain 54.5-61° W, 11-16° N for the hourly time steps closest to the ATR flights are considered. For the ATR, the median value for each flight (RF03-RF20) are considered. The number in the upper right corner of each panel represents the Pearson correlation coefficient between the respective $x$ and $y$ variable, and the dotted line the linear regression line (red for the COSMO$_{iso,1km}$ data and black for the ATR data).

1.6 L325: unclear sentence. Do you mean "... is assessed, without upscaling outputs from the higher resolution simulations to the lowest resolution"?
Yes, true, changed as suggested.

1.7 L418: "derive from it" → "conclude"? Done.

1.8 L420: "A behavior also" → "This behavior is also" Done.

1.9 L462: "10‰" → "10‰ in $\delta^2$H" Done.

1.10 L464: "promoting too" → "promoted by too"? I understand from section 3.1 that the number of cloud droplets and their size is determined by some tunable parameter, so the overestimated evaporation is a consequence of this, rather than the reverse?

*This is an interesting question. Our thinking here was rather that the evaporation rate is set by the dryness of the cloud top environment rather than by the cloud droplet size distribution. As shown in the profile Fig. 9, the RH and $q$ between 750 and 800 hPa is negatively biased in the simulations. A too dry cloud top environment can lead to too strong cloud top evaporation. We have adapted the text it in the following way:*

*"(2) Another possible explanation is that too strong cloud top evaporation occurs due to too strong turbulent mixing at the trade inversion. This could lead to only weak moistening but strong enrichment due to close to total (thus non-fractionating) droplet evaporation."*

1.11 L460-467: What is the difference between the two mechanisms? It sounds like they are equivalent. In both cases, overestimated cloud evaporation leads to weak moistening and strong enrichment. The underestimated precipitation efficiency in (1) sounds like just a bulk view to simply formalize this effect, whereas (2) is the detailed mechanism?

*It is true that both mechanisms can be seen as affecting precipitation efficiency. Thank you for pointing this out. We adapted the text as follows to make clear that in mechanism 1 we mean the conversion efficiency (from cloud droplets to sedimenting rain) in the cloudy updrafts within the cloud and with mechanism 2 we mean cloud water evaporation in the outflow region at cloud top:*

*"Too small conversion efficiencies (i.e., too low autoconversion rate within the cloud column) could explain an enrichment bias of about 10‰ in $\delta^2 H$ in the cloud top outflow region of shallow cumulus clouds. [...] (2) Another possible explanation is that too strong cloud top evaporation occurs due to too strong turbulent mixing at the trade inversion. This could lead to only weak moistening but strong enrichment due to close to total (thus non-fractionating) droplet evaporation."*

1.12 L477-479: The impact of the low-level dry bias on the high bias in d-excess could easily be quantified: the slope of d-excess as a function of surface relative humidity is easy to estimate under the Merlivat and Jouzel 1979 approximation. If the low-level dry bias is enough to explain the high bias in d-excess, then it is not necessary to invoke more complicated explanations.

*In the paper, we already formulated this hypothesis when discussing the d-excess bias:*

*"(2) Sea surface temperatures in $COSMO_{iso}$ are identical to the values from ECHAM6-wiso. Therefore, these alone should not be the origin of the bias. (3) However, in combination with the low-level dry bias in $COSMO_{iso}$, they result in stronger near-surface humidity gradients, which certainly contribute to the high bias in d-excess." Motivated by your comment, we now have looked into this in more detail using surface observations from the research vessel Meteor (Fig. R3). For this comparison, we extracted the d-excess and humidity values from the lowest model level. The humidity values were then used to calculate relative humidity with respect to sea surface temperature ($RH_{SST}$) in the model. The Meteor isotope observations (Galewsky, 2020) and $RH_{SST}$ (unpublished data provided by Sebastian Los, University of New Mexico; technical details described in Deutscher Wetterdienst (DWD), 2017) were collected in the context of EUREC[4]A. For the following analysis, only time steps at least 10 minutes before and 3 hours or more after measured rain are taken into account, since the Meteor isotope data was found to be heavily influenced by rainfall (Fig. S7 in Supplement of Bailey et al., 2023).*

*As shown in Fig. R3a, the data points of the simulated and observed datasets are distributed along the line that illustrates the d-excess vs. $RH_{SST}$ relationship with the closure assumption and the non-equilibrium fractionation factor from Merlivat and Jouzel (1979) in the smooth regime (at a wind speed of $6\,\mathrm{m\,s^{-1}}$). The differences between simulated and observed mean values are $-8\,\%$ for $RH_{SST}$ and $3.7\,‰$ for d-excess. From the Merlivat and Jouzel (1979) approximation, it follows that the $RH_{SST}$ bias accounts for a d-excess bias of $3.4\,‰$. In other words, the model bias in the d-excess at the surface is explained to a large extent with the bias in $RH_{SST}$. Note, that the d-excess bias in the lower troposphere (1000-850 hPa) is about $4.5\,‰$ (Table 2 of the paper), due to a stronger decrease of d-excess with altitude in the ATR data compared to the simulations (Fig. R3b). Consequently, we have adapted the discussion of the d-excess in Sect. 3.2 as follows:*

*"Possible reasons for a d-excess bias (Fig. 9h) are too strong near-surface humidity gradients or the use of inappropriate non-equilibrium fractionation factors. A comparison to ship-based observations from the Meteor during EUREC[4]A (not shown) revealed that in our case, too low relative humidity with respect to sea surface temperatures accounts for a large part of the identified d-excess bias. Since the sea surface temperatures in $COSMO_{iso}$ are identical to the ones in ECHAM6-wiso,*

*the too low relative humidity must result from the dry bias discussed above. The remaining share of the d-excess bias might be due to the choice of the non-equilibrium fractionation factor in the model simulations. Evidence is available that the chosen value in $COSMO_{iso}$ can still be improved compared to observations (Zannoni et al., 2022). The wind-dependent formulation of Merlivat and Jouzel (1979), used in ECHAM6-wiso, and the output from a sensitivity experiment with $COSMO_{iso}$ with a new wind-dependent formulation of the non-equilibrium fractionation factor from work done by Dütsch (2021, see experimental simulation shown in Supplement 4) yield d-excess values closer to the ATR observations than the evaluated $COSMO_{iso}$ simulations."*

In addition, comments regarding the d-excess bias in Sect. 5 were adapted.

[Figure]

**Figure R3: (a)** Relation between relative humidity with respect to sea surface temperature and surface d-excess for $COSMO_{iso,10km}$ (teal), $COSMO_{iso,5km}$ (yellow), $COSMO_{iso,1km}$ (red), and observations collected on the research vessel Meteor (gray) during the period 20 January to 13 February 2020. For the $COSMO_{iso}$ simulations, hourly mean values (small markers) and the mean value of the whole period (large markers) of the data in the domain 54.5-61° W, 11-16° N are shown. For the Meteor, observations at a temporal resolution of 1 minute (small markers) and the mean value of the whole period (large marker) are shown. During the considered period, the Meteor sampled in the domain 56.1-59.3° W, 11.6- 14.5° N. The approximation by Merlivat and Jouzel (1979) is illustrated by the black dashed line. **(b)** A zoom of the profiles shown in Fig. 9h of the paper together with the mean values (markers) from panel (a).

1.13 L636: the role of "partial evaporation of falling hydrometeors" is discussed, and then the conclusion is on "mesoscale transport processes". Doesn't the variability in d-excess rather reflect a combination of microphysical and mesoscale transport processes?

Yes, true, we adapted the text as follows:

*"In other words, the observed contrasts in d-excess between cloudy and dry-warm environments at cloud base, similar to q and $\delta^2H$, reflect mesoscale transport as well as the microphysical processes embedded in this flow."*

1.14 L668: "the higher the" → "at higher"? Done.

1.15 L706: "linked to shallow convective upward transport": and partial evaporation of falling hydrometeors?
Yes, true, changed as suggested.

1.16 Fig 5:

- To what extent is the simulated cloud-base cloud fraction sensitive to the threshold on condensed water content?
COSMO$_{iso}$ shows no sensitivity of cloud fraction to the chosen threshold in condensed water content. The cloud fraction at any level from 1000 to 400 hPa in COSMO$_{iso}$ is not sensitive to if the cloud water content (CWC) threshold (that must be exceeded for the identification of a cloudy grid point) is set to 0 or to $10 \, \text{mg kg}^{-1}$ (Fig. R5a). This is in contrast to ERA5 or ECHAM6-wiso, which both show a dependency on the chosen CWC threshold and overall show higher cloud fractions (black and purple in Fig. R5a). By looking at the CWC distribution of the cloudy grid points (combining the fraction of cloudy grid points in Fig. R5a with the CWC distribution of these cloudy grid points in Fig. R5b), we see that the smaller cloud fraction in COSMO$_{iso}$ is associated with higher CWC in clouds than in ERA5 or ECHAM-6wiso. In other words, the CW is more spatially concentrated in COSMO$_{iso}$ than in ERA5 or ECHAM-6wiso, which is due to the higher resolution of the COSMO$_{iso}$ simulations.

[Figure]

**Figure R5:** Vertical profiles of (a) cloud fraction and (b) cloud water content (CWC, only cloud liquid water, no rain water) from COSMO$_{iso,10km}$ (teal), COSMO$_{iso,5km}$ (yellow), COSMO$_{iso,1km}$ (red), ECHAM6-wiso (purple dashed), and ERA5 (black dotted). For each dataset, 35 profiles are randomly selected (to keep the statistics between the datasets comparable) in the domain 54.5-61° W, 11-16° N for each hourly time step that is closest to the time of the 18 ATR flights (RF03-RF20; 630 profiles in total). The number of grid points in the considered domain are 154 for ERA5, 3221 for COSMO$_{iso,10km}$, 12632 for COSMO$_{iso,5km}$, and 316028 for COSMO$_{iso,1km}$. The cloud fraction in (a) is calculated for every model level as the fraction of grid points (630 in total corresponding to the extracted profiles) with CWC exceeding $0 \, \text{mg kg}^{-1}$ (shading) or $10 \, \text{mg kg}^{-1}$ (lines). (b) The median (line) and 25-75-percentile range (shading) are computed taking into account only the grid points with CWC exceeding $10 \, \text{mg kg}^{-1}$. This figure is taken from Villiger (2022; their Fig. 6.4).

- Is there a possibility that the cloud fraction in observations and in the simulations represent two different things?

Indeed, the definition of the cloud fraction is ambiguous in observations, because clouds can be defined in multiple ways: as bodies that interact with radiation, or as objects that result from phase changes, or as ensembles of cloud droplets. However, in Bony et al. (2022), it is shown that these different definitions converge relatively well and that the time variations of the cloud fraction defined in these different ways were highly correlated, suggesting that they characterize the cloud field consistently. The definition we use here for clouds in the simulations (based on a cloud liquid water threshold; see description below) is relatively close to the definition of clouds in the ATR observations (see description below), suggesting that the comparison between simulations and observations is relevant. The procedures to determine cloud fraction in the numerical simulations vs. observations are as follows:

  – In the COSMO$_{iso}$ simulations, we identify a grid point as cloudy, if the cloud liquid water of the grid point exceeds $10 \, \text{mg kg}^{-1}$ (similar to Vial et al., 2019). The cloud fraction is then

calculated as the fraction of cloudy grid points in the domain 54.5-61° W, 11-16° N. The chosen domain is considerably larger than the area sampled by the ATR. However, with this domain, which represents the largest common area of the three COSMO$_{iso}$ simulations (i.e., roughly the COSMO$_{iso,1km}$ domain), we want to ensure a robust estimate of cloud fraction. Furthermore, we assume that the ATR observations are a representative sample of this area.

– The ATR cloud base cloud fraction used in the paper is based on a combination of horizontal radar and lidar scans (complete algorithm explained in Bony et al., 2022; summarized in their Fig. 12) that are pieced together to get a full scan of the cloud base rectangle (120 km × 20 km). The radar data has a spatial resolution of 12.5 m (within a 0-200 m radius around the aircraft) and 25 m (within a 200 m-12 km radius around the aircraft) and a temporal resolution of 1.5 s. The lidar data has a spatial resolution of 15 m (in a range of about 8 km around the aircraft) and a temporal resolution of 5 s. The combined radar-lidar product yields a classification of pixels into clear-sky, cloud, drizzle, and rain. Note that these classifications were quality-checked by comparing them with the cloud, drizzle and rain masks derived from the microphysical measurements conducted onboard the ATR (in which a cloud is detected if the liquid water content exceeds 10 mg kg$^{-1}$ and the water particle diameter lies between 1 and 100$\mu$m). The cloud fraction is then calculated as the area fraction of the cloud base rectangle covered by clouds; and the cloud+drizzle fraction (also shown in Fig. 5 of the paper because COSMO$_{iso}$ only distinguishes between cloud and rain water and it is unclear to which of the two categories drizzle should be compared) as the area fraction of the cloud base rectangle covered by clouds or drizzle.

Considering the differences outlined above, we do not claim that simulated and observed cloud fractions represent the 'same thing'. But we think that the comparison shown in Fig. 5 is still meaningful since it shows that, with a high enough resolution (i.e., 1km), the simulated cloud fractions lie within the range of the uncertainties of the observations.

- Is it possible that the sensitivity of the cloud fraction to the resolution is just an artifact of the different resolutions? Cloud fraction can be very sensitive to the resolution at which a cloudy pixel is defined. If all simulation outputs were coarse-grained on the same grid, would they exhibit such a large difference in cloud fraction? In other words, is the large difference related to different physical states, or to different definitions of a cloud? What does the observed cloud fraction represent, in terms of horizontal resolution of what is defined a cloud?

Absolutely. The sensitivity of the simulated cloud fractions to the spatial resolution is certainly an artefact of the different resolutions (and not the manifestation of different physical states). We expect that the cloud fraction is resolution-dependent, as long as the resolution is coarser than the clouds. Fig. R6 (which is taken from Supplement 1 of the paper) illustrates these mechanisms. If all simulations were aggregated to the same grid, we would expect very similar cloud fractions.

[Figure]

**Figure R6:** Schematic illustrating the strong dependence between cloud fraction and the spatial resolutions of the model. If the objects of interest (e.g., cloud, rain) are smaller than the grid resolution, the fraction of grid points with such an object decreases with increasing horizontal resolution.

Thanks to your comments, we realized that our text discussing Fig. 5 might be misleading. To

clarify our motivation for showing cloud fraction from all three simulations, we added the following sentence to Sect. 3.1:
*"The motivation for displaying cloud fractions from all three simulations is to assess at which resolution we obtain cloud fractions that approximate the observations within their uncertainty range."*
Additionally, we repeat that a resolution dependency is expected for cloud fraction from Sect. 3.1 in Sect. 5 (Summary and conclusion) as:
*"[...] cloud fraction at cloud base are overestimated but approach the observed values at higher spatial model resolution (for cloud fraction, this behaviour is expected [...]".*
* * *
**Reviewer 2:**

Overview: This is a review of "Water isotope characteristics of the cloud-circulation coupling in the North-Atlantic trades. Part 1: A process-oriented evaluation of COSMO$_{iso}$ simulations with EUREC4A observations" by Villiger et al.. This paper is part of a multi-part paper of which I have only reviewed Part 1. This paper is an evaluation of COSMO simulations that are to be used to interpret the mechanisms behind trade wind cumulus.
I find the paper and the methods interesting, but there are some things that can be improved, therefore I suggest major revisions.
First of all, I would like to comment that the paper is thoroughly written, it is clear what has been done and what the results and interpretation of the experiments are. Also, I appreciate that the authors have made nice figures with a consistent layout and sizing throughout the paper, showing that the paper has been made with care.
We thank the reviewer very much for the positive feedback about the quality of the writing and figures. We appreciate their effort to thoroughly and critically read this (admittedly) long paper a lot.

**Major comments:**

A Despite the results being clearly explained, I did not find it easy to extract the true motivation of the paper. From the title I expected the paper to be primarily about validating the model for isotopes studies, but in the end it seems to be more about evaluating COSMO$_{iso}$ for trade wind cumulus simulations. Maybe the whole study could benefit from reducing this paper largely in size and merge it with the other Part 2 paper, in order to have a very original isotopes paper.
We acknowledge the need for more motivation for the use of water isotopes in the introduction to help a non-water-isotope readership to understand the need for this first part of the paper (see minor comment 2.4 below). Evaluating isotope-enabled simulations of shallow cumulus cloud formation in varying situations of large-scale flow entails investigating the key physical variables controlling isotope variability in a first indispensable step. This first step is not trivial because it requires isotope observations in shallow cumulus clouds and their environment for a series of different large-scale flow situations. We have collected these unique observations from about 80 hours of flight during EUREC$^4$A (Bony et al., 2022; Bailey et al., 2023) and combine them in this study with regional isotope-enabled simulations at three different horizontal resolutions. These are the first isotope-enabled simulations at such a high resolution in a realistic meteorology setup. In this paper we thus combine two novel and very valuable observational and model datasets. Therefore, the evaluation step of bringing these two datasets together by comparing key statistics of the measured and observed isotope signals and the variables driving isotope variability is crucial and should be done thoroughly before reverting the perspective in part 2 and using isotopes as tracers for the mesoscale cloud-relative overturning circulation and phase change processes embedded in shallow cumulus clouds.
As discussed in the answer to minor comment 1.1 of Reviewer 1, we understand the remark about the length of part 1. As recommended by Reviewer 1 (and Reviewer 2 of part 2) we have balanced the lengths and the level of detail in the writing of the two parts. We have shortened part 1 by leaving out the analyses about the subcloud layer (see answer to comment 1.1. above). Given the importance of the evaluation of this first series of isotope-enabled simulations for tropical low clouds with an unprecedented set of isotope observations, as well as the novelty of the approach used in the feature-based analysis in the section "Drivers of the horizontal variability of isotopes at cloud base" we do not agree with the idea of a fusion of parts I and II.

B One thing that I found particularly striking, and maybe even worrying is that the authors state

that the ultimate goal is to use isotopes and simulations to interpret the cloud system better, but at the same time, they conclude that while precipitation, cloud fraction and liquid water content are sensitive to grid spacing, temperature, humidity, and isotopes are not. If this is the case, how can isotopes then ultimately be of help in learning about cloud processes?

We regret that our paper seems to leave the impression that isotopes are not useful to learn more about shallow cloud formation and organisation in various large-scale flow situations. We think that this impression arises from a misunderstanding about the mentioned result that some cloud variables are resolution-dependent, while temperature, humidity and isotopes are not. We interpret this result in an entirely different way. The fact that a resolution-dependence in the distribution or percentile of a particular cloud variable is observed and not in isotope variables, does not imply that isotopes are not impacted by cloud formation. It just means that the overall distribution of isotope variables, which is dominated by clear-sky areas in the trades, is not systematically impacted by cloudiness in the selected region.

The reason for the resolution-dependence of cloud-related variables is explained in detail in Supplement 1. As opposed to the highly skewed cloud variables, the water isotopes, humidity and temperature are close-to-normally distributed, and in the traditional evaluation part of the paper (Section 3) we compare their full distribution across the cloud base environment (cloudy and cloud free) between model simulations and observations. We observe a weak resolution dependence, when comparing the isotope signals ($\delta^2$H and d-excess) associated with clouds and dry-warm patches (Fig. R7).

We clarify this aspect of resolution dependence at the beginning of Section 3.2:

*"In contrast to the highly skewed variables discussed in the previous section, the close-to-normally distributed variables considered in this section are largely indifferent to model resolution (Fig. 9 and 10). This limited resolution dependence results from the fact that here we consider the full distribution of these variables and do not focus on the properties of subgrid-scale features as in the previous section."*

In addition, as suggested by the reviewer in their comment 2.4, we added some more background on our motivation for using isotopes in the introduction. Thank you for pointing us towards this potential for misunderstanding.

Also, if cloud processes are key, wouldn't it make more sense to do high-resolution large-eddy simulations with isotopes, as these capture the cloud processes much better? I would like to ask the authors to clarify the motivation and to convince the reader better that their goal has chance in succeeding.

We agree that if we were only interested in cloud processes, then LES would be the best choice. But here we are interested in how processes across a range of scales interact to form shallow cumulus clouds. We know from an earlier study (Aemisegger et al., 2021) that the large-scale flow strongly impacts the organisation of shallow clouds and that isotope observations from the subcloud layer reflect the transport history of the air masses. Therefore, our choice of the best possible model setup is based on a trade-off between resolution and domain size. We would like to have an as small as possible grid-spacing to represent shallow cumuli realistically, while keeping the domain size as large as possible to have a realistic representation of the synoptic- and meso-scale meteorology. The domain size matters in particular for adopting the Lagrangian perspective in part 2, in which we compute trajectories several days back in time to study the transport history of the air parcels reaching the trades. The setup we chose with the regional isotope-enabled COSMO$_{\text{iso}}$ model is well-suited for our aim to use isotopes as tracers for the interaction of shallow clouds with the large-scale flow. Apart from COSMO$_{\text{iso}}$, there are not many isotope-enabled models for high-resolution studies available: there is isotope-enabled SAM (Blossey et al., 2010) and, isotope-enabled WRF (Moore et al., 2016). However, to the best of our knowledge, isotope-enabled SAM has been used in radiative-convective equilibrium only and not in realistic meteorology setups with boundary data from a global isotope-enabled model simulation. To accommodate for the reviewer's remark, we added the following motivation sentence in the introduction to justify our approach:

*"This hypothesis can only be addressed with multi-scale, regional model simulations with realistic, time-dependent boundary conditions, which provide a statistically robust, three-dimensional picture of cloud characteristics and associated isotope signals. In addition, the EUREC⁴A ATR observational isotope dataset is essential for evaluating the COSMO$_{\text{iso}}$ simulation data before using it to*

[Figure]

**Figure R7:** Distributions of cloud base **(a)** specific humidity, **(b)** $\delta^2$H in vapour, and **(c)** d-excess in vapour of the data points identified as *dry-warm*, *clear*, *cloud*, and *cloud-rain* in the domain 54.5-61° W, 11-16° N over the whole simulated period (20 January to 13 February 2020). Shown are the median (markers), the 25-75-percentile range (lines) for the three COSMO$_{iso}$ simulations. **(d)** Percentage of cloud base grid points in each category.

*deepen our process understanding.*"

C   Also, the authors present suggestions for improvement in the conclusion that in my view should be conducted as the suggested improvements would largely increase the qualities of the simulations and the work does not look that extensive.

We appreciate the recommendation of the reviewer, but we cannot include any further model simulations for reasons of lacking resources (both computational and in personnel for their evaluation) and in the interest of keeping the manuscript short. There is no reason to think that the recommendations for additional simulations, provided in the conclusions, will indeed solve all the issues observed. Furthermore, we point the reviewer to a series of other simulations with other regional (Beucher et al., 2022) or even global high-resolution models (Schulz and Stevens, 2023), which show similar biases (cold-dry bias) as we find here.

To clarify this, we have removed the final sentence of the abstract (*"Additionally, we provide explicit recommendations for adaptations of the modelling setup to be tested in future research."*) to avoid the unintended impression of a model-development study.

**Minor comments:**

2.1 Line 1: Why do isotopes have this potential? Please state this

We added the following sentence at the beginning of the abstract:

*"Naturally available, stable and heavy water molecules such as HDO and $H_2^{18}O$ have a lower saturation vapour pressure than the most abundant light water molecule $H_2^{16}O$ and therefore these heavy water molecules preferentially condense and rain out during cloud formation. Stable water isotope observations thus have the potential to provide information on cloud processes in the trade-wind region, in particular when combined with high-resolution model simulations."*

The difference in saturation vapour pressure is a first order control on the isotopic composition of atmospheric waters. In addition, during non-equilibrium processes such as surface evaporation or partial evaporation of rain drops, the difference in diffusivity between the heavy and the light molecule comes into play. We now clarify this in the introduction.

2.2 Line 10: Doesn't this statement show that the simulations will not help much in interpreting the isotope measurements?

- If the reviewer means here the part about the resolution dependence of cloud variables vs. the resolution independence of isotope, humidity and temperature: As explained in our answer to major comment B above, we disagree with this interpretation. We changed the text as follows to avoid rising such a misunderstanding in our readership:
  *"Cloud fraction and liquid water content show a better agreement with aircraft observations with higher spatial resolution because they show strong spatial variations on the scale of a few kilometres."*

- If the reviewer means that the biases in isotope variables implies that the simulations will not help much in interpreting the measurements: in our opinion and as we show in section 4 using an anomaly-based approach, a bias in a simulated variable does not imply that the model is not useful for process studies and to provide a spatial context to the observations.

2.3 Line 22-25: I do not understand how the results stated before lead to the conclusion that COSMO_iso evaluation suggests that the simulations can be used for studying the impact of circulation on clouds, as cloud properties do not converge with resolution.

On this point, see also our answer to major comment B. Here, we assume that the reviewer specifically means cloud fraction and our Fig. 5 (for a discussion on precipitation, we refer to comment 2.8 below). We did not design this series of simulations with the aim of reaching convergence in cloud properties, but with the aim of going as high as necessary in resolution for reaching a satisfactory comparison with the aircraft-based observations. We needed the 10 and 5 km simulations in our nested simulation approach to obtain a robust setup for the 1 km simulation, in which we reached a satisfactory representation of cloud fraction compared to observations, when keeping in mind the uncertainties associated with cloud base cloud fraction estimates from observations (0.03-0.08%, see Fig. 18b in Bony et al., 2022).

In addition, we have reformulated parts of Sect. 5 (Summary and conclusion) to explicitly state how we get to this conclusion:

*"The motivation for the evaluation of the three $COSMO_{iso}$ simulations was to assess, whether they can be used to study the role of the atmospheric circulation at different scales for controlling the formation of shallow cumulus clouds in the trades using isotopes. The realistic representation of cloud organisation patterns, cloud fraction, precipitation regimes and the temporal evolution of isotope variables at cloud base suggest that $COSMO_{iso,1km}$ can serve this purpose, while the simulations with coarser resolution are still useful for looking at the role of the larger-scale circulation as a driver for shallow cloud organisation. The identified model biases in humidity, temperature, and isotope signals in vapour are of minor concern because they can be eliminated by looking at anomalies, as shown in the feature-based evaluation (distinguishing between different environments named dry-warm, clear, cloud, and cloud-rain) of $COSMO_{iso,1km}$. The feature-based evaluation revealed that the low-level biases in the $COSMO_{iso}$ simulations discussed above are not feature-specific but*

*apply to all the different cloud base environments (cf. Supplement 3). In other words, the processes linked to the formation of these features are accurately reproduced by COSMO$_{iso}$."*

2.4 Introduction: the introduction is lacking hypotheses how isotopes could help in improving understanding. It would help the reader (I am a reader that knows very little about isotopes) to state how isotope measurements in combination with simulations could show something that the simulations, nor the data analysis of isotopes alone could not.
We added two hypotheses to the introduction, which shall illustrate the value of using a combination of observations and simulations to fulfil the aims of the two-part paper:
*"Specifically, we test the hypothesis that isotopes are modulated by both microphysical processes in the cloud-relative overturning circulation and variations in the large-scale flow. In turn, this means that isotopes have the potential to provide an observational constraint on these processes that are otherwise very difficult to observe. This hypothesis can only be addressed with multi-scale, regional model simulations with realistic, time-dependent boundary conditions, which provide a statistically robust, three-dimensional picture of cloud characteristics and associated isotope signals. In addition, the EUREC$^4$A ATR observational isotope dataset is essential for evaluating the COSMO$_{iso}$ simulation data before using it to deepen our process understanding. For this evaluation, three objectives are pursued: [...]"*

2.5 Line 204: The cited papers from Seifert & Beheng describe a 2 moment scheme. Please clarify.
Indeed, the references to Seifert & Beheng were confusing, they were actually related to the chosen cloud droplet number. We removed these references.

2.6 Line 206: How does one explicitly resolve shallow convection at grid spacings larger than the size of a shallow cloud?
Shallow convection is only partly resolved, part of it is taken over by the turbulence scheme. We adapted the text accordingly:
*"The convection schemes of the model (Tiedtke, 1989; Theunert and Seifert, 2006) were disabled, meaning that deep and shallow convection were treated explicitly at the grid scale."*
We have used the COSMO$_{iso}$ model in this setup, i.e. with shallow and deep convection parametrisations switched off already in many previous studies (Dahinden et al., 2021; Diekmann et al., 2021; Thurnherr et al., 2021; De Vries et al., 2022), in which the comparison with isotope observations in various regions of the world have shown a good performance of the explicit convection setup. Furthermore, as outlined in the introduction, previous analyses have shown that COSMO simulations at a range of resolutions (grid spacing $\leq$ 25 km) do not necessarily provide more realistic results in terms of radiation and precipitation patterns if the parameterization of shallow convection is turned on (Vergara-Temprado et al., 2019).

2.7 Section 2.2 is very well written and explains the data collection very well.
Thank you for this positive feedback.

2.8 Line 410: What is your opinion on the quality of the precipitation distribution? Is it good enough? Should we expect further improvements if we move to LES resolution?
Here we can only speculate, since performing realistic isotope-enabled large domain LES simulations for EUREC$^4$A cases is much beyond the scope of this work and has not been done, to the best of our knowledge, in any other study. Furthermore, we would like to emphasise that the definition of what is good enough in terms of the precipitation distribution is not a priori clear (see e.g. Wernli et al., 2008 for a discussion about the evaluation of precipitation forecasts). LES simulations might perform better in representing precipitation statistics at individual grid points, however not necessarily in terms of spatial organisation. The representation of trade-cumulus precipitation among LES differs largely, as shown in vanZanten et al. (2011). For LES simulations and their evaluation in terms of precipitation patterns and formation processes for EUREC$^4$A cases, we refer to Schulz and Stevens (2023) and Radtke et al. (2023). We added a reference to these ICON-LES simulations in the text. Furthermore, note that an inter-comparison between convection-resolving model and LES simulations with EUREC4A data is ongoing in the context of EUREC$^4$A-MIP (https://eurec4a.eu/motivation)).

2.9 Section 4.1: One open question connecting to my main comments is how well do we understand isotopic processes? Are there indications that the results presented here also teach us something new about isotopic fractionation itself?
Thank you for this important comment. Isotope meteorology is a relatively young discipline, with

novel observational capabilities having arisen in the last decade with laser absorption-based measurement techniques allowing for high-quality in-situ observations (see, e.g. Aemisegger et al., 2012 and Galewsky et al., 2016). These instruments have allowed us to investigate isotope signals at much shorter timescales than was previously possible. This has given rise to novel studies, such as this one, on the interpretation of isotope signals at the timescale of cloud formation. In terms of isotope fractionation our understanding is very good with respect to equilibrium vapour-liquid fractionation, which is well-understood from a theoretical (Bigleisen, 1961) and empirical (e.g., Majoube, 1971; Horita and Wesolowski, 1994) point of view. However, with respect to non-equilibrium fractionation, occurring in under- or super-saturated conditions, there are still uncertainties in the formulation of the fractionation factor (e.g. Pfahl and Wernli, 2009; Zannoni et al., 2022). In Supplement 4, a sensitivity test is presented on this topic of active research with a novel formulation for the non-equilibrium fractionation factor proposed by Marina Dütsch in a paper in preparation (Dütsch et al., 2023).

2.10 Conclusions: I sometimes wonder if this separation in Flower, Sugar, Gravel, Fish is bringing so much progress to the field. It is a rather arbitrary classification and one could wonder if these are really the fundamental types. Do the isotope characteristics also vary significantly between the types?
Although these four patterns were indeed originally defined subjectively (based on visual analysis), and although it is well recognized that there is a continuum of cloud organizations in the trades, several studies have since shown that they actually represent 'extrema' in different cloud organization metrics (e.g., Bony et al., 2020; using an objective classification method). They also correspond to patterns that emerge in machine learning classifications (Denby 2020; Rasp et al., 2020) or in principal component analyses (Janssens et al., 2021). Furthermore, the four cloud patterns are associated with different radiative signatures (e.g., Bony et al., 2020) and relate to distinct large-scale flow regimes (Aemisegger et al., 2021; Schulz et al., 2021). A change in the large-scale flow regimes over the North Atlantic with global warming would therefore likely lead to a change in the occurrence frequency of cloud organisation patterns in the trades, thereby impacting the radiative feedback of this region. The isotope characteristics associated with the different predominant flow regimes shaping the major cloud organisation patterns have been analysed in detail in a previous study (Aemisegger et al., 2021). Lastly, perhaps even more importantly for the success of a large international field campaign like EUREC[4]A, these cloud patterns helped a lot with the communication in the community during the campaign and now in papers. We interpret these cloud patterns as extreme cases of a continuum of cloud patch sizes and shapes.

2.11 Line 685: should this sensitivity analysis not be part of this paper? The goal is to let COSMO reproduce the case as good as possible, and there are clear deficiencies.
It is not a priori clear if these additional simulations and sensitivity experiments would provide much better results. We have mentioned these ideas to encourage future model-based research on shallow cumulus clouds with water isotopes as tracers. For our simulations, we have chosen the currently best available isotope-enable regional model setup to reproduce the meteorology and isotope variability during EUREC[4]A. The performance of our simulations is comparable in many aspects to other model simulations of EUREC[4]A cases (Beucher et al., 2022; Schulz and Stevens, 2023). The fact that there are systematic deviations in temperature and humidity profiles (lower tropospheric cold-dry bias) is thus not model-specific. And since our paper is focused on process understanding and is not a model development study, in which case we would have submitted it to the journal Geophysical Model Development, we do not aim at solving this complex issue.

2.12 Line 695: Same as the previous comment. Please try this, as you need high quality simulations for your purpose.
Same answer as above.

**References**

Aemisegger, F., Vogel, R., Graf, P., Dahinden, F., Villiger, L., Jansen, F., Bony, S., Stevens, B., and Wernli, H.: How Rossby wave breaking modulates the water cycle in the North Atlantic trade wind region. Weather Clim. Dynam., 2, 281–309, `https://doi.org/10.5194/wcd-2-281-2021`, 2021.

Aemisegger, F., Sturm, P., Graf, P., Sodemann, H., Pfahl, S., Knohl, A., and Wernli, H.: Measuring variations of $\delta^{18}O$ and $\delta^2H$ in atmospheric water vapour using two commercial laser-based spectrometers: an instrument characterisation study, Atmos. Meas. Tech., 5, 1491–1511, `https://doi.org/10.5194/amt-5-1491-2012`, 2012.

Albright, A. L., Bony, S., Stevens, B., and Vogel, R.: Observed subcloud layer moisture and heat budgets in the trades. J. Atmos. Sci., 79, 2363–2385. `https://doi.org/10.1175/JAS-D-21-0337.1`, 2022.

Bailey, A., Aemisegger, F., Villiger, L., Los, S. A., Reverdin, G., Quiñones Meléndez, E., Acquistapace, C., Baranowski, D. B., Böck, T., Bony, S., Bordsdorff, T., Coffman, D., de Szoeke, S. P., Diekmann, C. J., Dütsch, M., Ertl, B., Galewsky, J., Henze, D., Makuch, P., Noone, D., Quinn, P. K., Rösch, M., Schneider, A., Schneider, M., Speich, S., Stevens, B., and Thompson, E. J.: Isotopic measurements in water vapor, precipitation, and seawater during EUREC4A. Earth Syst. Sci. Data, 15, 465–495, `https://doi.org/10.5194/essd-15-465-2023`, 2023.

Beucher, F., Couvreux, F., Bouniol, D., Faure, G., Favot, F., Dauhut, T., and Ayet, A.: Process oriented evaluation of the oversea AROME configuration: focus on the representation of cloud organisation. Q. J. Roy. Meteor. Soc., 2–37, `https://doi.org/10.1002/qj.4354`, 2022.

Bigleisen, J.: Statistical mechanics of isotope effects on the thermodynamic properties of condensed systems. J. Chem. Phys., 34, 1485–1493, `https://doi.org/10.1063/1.1701033`, 1961.

Blossey, P. N., Kuang, Z., and Romps, D. M.: Isotopic composition of water in the tropical tropopause layer in cloud-resolving simulations of an idealized tropical circulation. J. Geophys. Res., 115, D24309, `https://doi.org/doi:10.1029/2010JD014554`, 2010.

Bony, S., Schulz, H., Vial, J., and Stevens, B.: Sugar, gravel, fish and flowers: Dependence of mesoscale patterns of trade-wind clouds on environmental conditions. Geophys. Res. Lett., 47, e2019GL085988. `https://doi.org/10.1029/2019GL085988`, 2020.

Bony, S., Lothon, M., Delanoë, J., Coutris, P., Etienne, J.-C., Aemisegger, F., Albright, A. L., André, T., Bellec, H., Baron, A., Bourdinot, J.-F., Brilouet, P.-E., Bourdon, A., Canonici, J.-C., Caudoux, C., Chazette, P., Cluzeau, M., Cornet, C., Desbios, J.-P., Duchanoy, D., Flamant, C., Fildier, B., Gourbeyre, C., Guiraud, L., Jiang, T., Lainard, C., Le Gac, C., Lendroit, C., Lernould, J., Perrin, T., Pouvesle, F., Richard, P., Rochetin, N., Salaün, K., Schwarzenboeck, A., Seurat, G., Stevens, B., Totems, J., Touzé-Peiffer, L., Vergez, G., Vial, J., Villiger, L., and Vogel, R.: EUREC4A observations from the SAFIRE ATR42 aircraft. Earth Syst. Sci. Data, 14, 2021–2064. `https://doi.org/10.5194/essd-14-2021-2022`, 2022.

Dahinden, F., Aemisegger, F., Wernli, H., Schneider, M., Diekmann, C. J., Ertl, B., Knippertz, P., Werner, M., and Pfahl, S.: Disentangling different moisture transport pathways over the eastern subtropical North Atlantic using multi-platform isotope observations and high-resolution numerical modelling. Atmos. Chem. Phys., 21, 16319–16347, `https://doi.org/10.5194/acp-21-16319-2021`, 2021.

Denby, L.: Discovering the importance of mesoscale cloud organization through unsupervised classification. Geophys. Res. Lett., 47, 1–10. `https://doi.org/10.1029/2019GL085190`, 2020.

Deutscher Wetterdienst (DWD): Bordwetterwarte FS Meteor. Technical report, `https://www.dwd.de/DE/fachnutzer/schifffahrt/maritimberatung/_functions/Teasergroup/thema_bordwetterdienst.html`, 2017.

de Vries, A. J., Aemisegger, F., Pfahl, S., and Wernli, H.: Stable water isotope signals in tropical ice

clouds in the West African monsoon simulated with a regional convection-permitting model. Atmos. Chem. Phys., 22, 8863–8895, `https://doi.org/10.5194/acp-22-8863-2022`, 2022.

Diekmann, C. J., Schneider, M., Knippertz, P., de Vries, A. J., Pfahl, S., Aemisegger, F., Dahinden, F., Ertl, B., Khosrawi, F., Wernli, H., Braesicke, P.: A Lagrangian perspective on stable water isotopes during the West African Monsoon. J. Geophys. Res., 126, e2021JD034895. `https://doi.org/10.1029/2021JD034895`, 2021.

Dütsch, M.: A new theoretical framework for parameterizing nonequilibrium fractionation during evaporation from the ocean [conference presentation]. Workshop, 15-17 November, Water Isotopes: From Weather to Climate. `https://youtu.be/Cd28fK_1TUs`.

Dütsch, M., Fairall, C. W., Fiorella, R. P., and Blossey, P. N. (2023). Parameterizing surface evaporation with moisture roughness lengths: Application to evaporation of water isotopologues. In Preparation for J. Adv. Model. Earth Sy.

Galewsky, J., Steen-Larsen, H. C., Field, R. D., Worden, J., Risi, C., and Schneider, M.: Stable isotopes in atmospheric water vapor and applications to the hydrologic cycle. Rev. Geophys., 54, 809–865, `https://doi.org/10.1002/2015RG000512`, 2016.

Galewsky, J.: Level-1 Continuous In-situ Water Vapor Isotopic Composition [data set]. AERIS. `https://doi.org/10.25326/83`, 2020.

Horita, J. and Wesolowski, D. J.: Liquid-vapour fractionation of oxygen and hydrogen isotopes of water from the freezing to the critical temperature. Geochim. Cosmochim. Ac., 58, 3425–3437, `https://doi.org/10.1016/0016-7037(94)90096-5`, 1994.

Janssens, M., Vilà-Guerau de Arellano, J., Scheffer, M., Antonissen, C., Siebesma, A. P., and Glassmeier, F.: Cloud patterns in the trades have four interpretable dimensions. Geophys. Res. Lett., 48, 1–11. `https://doi.org/10.1029/2020GL091001`, 2021.

Majoube, M.: Fractionnement en oxygène-18 et en deutérium entre l'eau et sa vapeur. J. Chim. Phys., 68, 1423–1436, `https://doi.org/10.1051/jcp/1971681423`, 1971.

Merlivat, L., and Jouzel, J.: Global climatic interpretation of the deuterium-oxygen 18 relationship for precipitation. J. Geophys. Res.-Oceans, 84, 5029–5030, `https://doi.org/10.1029/JC084iC08p05029`, 1979.

Moore, M., Blossey, P. N., Muhlbauer, A., and Kuang, Z.: Microphysical controls on the isotopic composition of wintertime orographic precipitation. J. Geophys. Res. Atmos., 121, 7235–7253, `https://doi.org/10.1002/2015JD023763`, 2016.

Radtke, J., Naumann, A.K., Hagen, M. and Ament, F.: The relationship between precipitation and its spatial pattern in the trades observed during EUREC4A. Q. J. Roy. Meteor. Soc., 148, 1913– 1928. Available from: `https://doi.org/10.1002/qj.4284`, 2022.

Rasp, S., Schulz, H., Bony, S., and Stevens, B.: Combining crowd-sourcing and deep learning to explore the meso-scale organization of shallow convection. B. Am. Meteorol. Soc., 2020, 1980–1995. `https://doi.org/10.1175/BAMS-D-19-0324.1`, 2020.

Schulz, H., Eastman, R., and Stevens, B.: Characterization and evolution of organized shallow convection in the trades. J. Geophys. Res.-Atmos., 126, e2021JD034575, `https://doi.org/10.1029/2021JD034575`, 2021.

Schulz, H., and Stevens, B.: On the representation of shallow convection in the trades by large-domain, hecto-meter, large-eddy simulations. submitted to J. Adv. Model. Earth Sy., `https://eartharxiv.org/repository/view/4986/`, 2023.

Stevens, B., Bony, S., Brogniez, H., et al. Sugar, gravel, fish and flowers: Mesoscale cloud patterns in the trade winds. Q. J. Roy. Meteor. Soc., 146: 141–152. `https://doi.org/10.1002/qj.3662`, 2020.

Thurnherr, I., Hartmuth, K., Jansing, L., Gehring, J., Boettcher, M., Gorodetskaya, I., Werner, M., Wernli, H., and Aemisegger, F.: The role of air–sea fluxes for the water vapour isotope signals in the cold and warm sectors of extratropical cyclones over the Southern Ocean. Weather Clim. Dynam., 2, 331–357, `https://doi.org/10.5194/wcd-2-331-2021`, 2021.

vanZanten, M. C., Stevens, B., Nuijens, L., Siebesma, A. P., Ackerman, A. S., Burnet, F., Cheng, A., Couvreux, F., Jiang, H., Khairoutdinov, M., Kogan, Y., Lewellen, D. C., Mechem, D., Nakamura, K., Noda, A., Shipway, B. J., Slawinska, J., Wang, S., and Wyszogrodzki, A.: Controls on precipitation and cloudiness in simulations of trade-wind cumulus as observed during RICO. J. Adv. Model. Earth Sy., 3, `https://doi.org/10.1029/2011MS000056`, 2011.

Vergara-Temprado, J., Ban, N., Panosetti, D., Schlemmer, L., and Schär, C.: Climate models permit convection at much coarser resolutions than previously considered. J. Climate, 33, 1915–1933, `https://doi.org/10.1175/JCLI-D-19-0286.1`, 2019.

Vial, J., Vogel, R., Bony, S., Stevens, B., Winker, D. M., Cai, X., Hohenegger, C., Naumann, A. K., and Brogniez, H.: A new look at the daily cycle of tradewind cumuli. J. Adv. Model. Earth Sy., 11, 3148–3166. `https://doi.org/10.1029/2019ms001746`, 2019.

Villiger, L.: Large-scale circulation drivers and stable water isotope characteristics of shallow clouds over the tropical North Atlantic. Doctoral Thesis, ETH Zürich, `https://doi.org/10.3929/ethz-b-000586270`, 2022.

Wernli, H., M. Paulat, M. Hagen, and Frei, C.: SAL—A Novel Quality Measure for the Verification of Quantitative Precipitation Forecasts. Mon. Wea. Rev., 136, 4470–4487, `https://doi.org/10.1175/2008MWR2415.1`, 2008.